# Model-Based Exploration in Monitored Markov Decision Processes

**Alireza Kazemipour** [1 2]   **Matthew E. Taylor** [1 2 3]   **Michael Bowling** [1 2 3]

## Abstract

A tenet of reinforcement learning is that the agent always observes rewards. However, this is not true in many realistic settings, e.g., a human observer may not always be available to provide rewards, sensors may be limited or malfunctioning, or rewards may be inaccessible during deployment. Monitored Markov decision processes (Mon-MDPs) have recently been proposed to model such settings. However, existing Mon-MDP algorithms have several limitations: they do not fully exploit the problem structure, cannot leverage a known monitor, lack worst-case guarantees for "unsolvable" Mon-MDPs without specific initialization, and offer only asymptotic convergence proofs. This paper makes three contributions. First, we introduce a model-based algorithm for Mon-MDPs that addresses these shortcomings. The algorithm employs two instances of model-based interval estimation: one to ensure that observable rewards are reliably captured, and another to learn a minimax-optimal policy. Second, we empirically demonstrate the advantages. We show faster convergence than prior algorithms in more than four dozen benchmarks, and even more dramatic improvements when the monitoring process is known. Third, we present the first finite sample-bound on performance. We show convergence to a minimax-optimal policy even when some rewards are never observable.

## 1. Introduction

Reinforcement learning (RL) is based on trial-and-error: instead of being directly shown what to do, an agent receives consistent numerical feedback in the form of rewards for its decisions. However, this assumption is not always re-

alistic because feedback often comes from an exogenous entity, such as humans (Shao et al., 2020; Hejna & Sadigh, 2024) or monitoring instruments (Vu et al., 2021). Assuming the reward is available at all times is not reasonable in such settings, e.g., due to human time constraints (Pilarski et al., 2011), hardware failure (Bossev et al., 2016; Dixit et al., 2021), or inaccessible rewards during deployment (Andrychowicz et al., 2020). Hence, relaxing the assumption that rewards are always observable would mark a significant step toward agents continually operating in the real world. As an extension of Markov decision processes (MDPs), monitored Markov decision processes (Mon-MDPs) (Parisi et al., 2024b) have been proposed to model such situations, although algorithms for Mon-MDPs remain in their infancy (Parisi et al., 2024a). Existing algorithms do not leverage the structure of Mon-MDPs, focusing exploration uniformly across the entire state-action space, and only have asymptotic guarantees without providing sample complexity bounds. Furthermore, they have focused on *solvable* Mon-MDPs, where observing every reward under some circumstances is possible. The original introduction of Mon-MDPs also considered *unsolvable* Mon-MDPs, proposing a minimax formulation as the optimal behavior, but no algorithms have explored this setting.

In this paper, we introduce *Monitored Model-Based Interval Estimation with Exploration Bonus (Monitored MBIE-EB)*, a model-based algorithm for Mon-MDPs that offers several advantages over previous algorithms. Monitored MBIE-EB exploits the Mon-MDP structure to consider the uncertainty on each unknown component separately. This approach also makes it the first algorithm that can take advantage of situations where the agent knows the monitoring process in advance. Furthermore, Monitored MBIE-EB balances optimism in uncertain quantities with pessimism for rewards that have never been observed, reaching the minimax-optimal behavior in unsolvable Mon-MDPs. The trade-off between optimism and pessimism is challenging because pessimism may dissuade agents from exploring sufficiently in solvable Mon-MDPs. We address this by having a *second* instance of MBIE-EB to force the agent to efficiently observe all rewards that can be observed. Building off of MBIE-EB (Strehl & Littman, 2008), we prove the first polynomial sample complexity bounds (Kakade, 2003; Lattimore & Hutter, 2012) as the measure of Monitored

[1]Department of Computing Science, University of Alberta [2]Alberta Machine Intelligence Institute (Amii) [3]Canada CIFAR AI Chair, Amii. Correspondence to: Alireza Kazemipour <kazemipo@ualberta.ca>.

*Proceedings of the 42^nd International Conference on Machine Learning*, Vancouver, Canada. PMLR 267, 2025. Copyright 2025 by the author(s).

MBIE-EB's efficiency, which applies equally to solvable and unsolvable Mon-MDPs. We then explore the Monitored MBIE-EB's efficacy empirically. We present its efficient exploration in practice, outperforming the recent Directed Exploration-Exploitation algorithm (Parisi et al., 2024a) in four dozen benchmarks, including all of environments from Parisi et al. (2024a). We also demonstrate that Monitored MBIE-EB converges to optimal policies in solvable Mon-MDPs and minimax-optimal policies in unsolvable Mon-MDPs. This confirms Monitored MBIE-EB is able to separate the solvable Mon-MDPs from the unsolvable ones. Finally, we illustrate Monitored MBIE-EB can exploit knowledge of the monitoring process to learn even faster.

## 2. Preliminaries

Traditionally, RL agent-environment interaction is modeled as a Markov decision process (MDP) (Puterman, 1994; Sutton & Barto, 2018): at every timestep $t$ the agent performs an action $A_t$[1], according to the environment state $S_t$; in turn, the environment transitions to a new state $S_{t+1}$ and generates a bounded numerical reward $R_{t+1}$. It is assumed that the agent always observes the rewards. Any partial observability is only considered for environment states, resulting in partially observable MDPs (POMDPs) (Kaelbling et al., 1998; Chadès et al., 2021). Until recently, prior work on partially observable rewards was limited to active RL (Schulze & Evans, 2018; Krueger et al., 2020), RL from human feedback (RLHF) (Kausik et al., 2024), options framework (Machado & Bowling, 2016), and reward-uncertain MDPs (Regan & Boutilier, 2010). However, these frameworks lack the complexity to formalize situations where the reward observability stems from some stochastic processes. In active RL, the reward can always be observed by simply paying a cost; in RLHF, the reward is either never observable (with no guidance coming from the human) or always observable (with the human providing all the guidance). Neither of these settings capture scenarios, where there are rewards that the agent can only *sometimes* see and whose observability can be predicted and possibly controlled.

Recently, inspired by partial monitoring (Bartók et al., 2014), Parisi et al. (2024b) extended the MDP framework to also consider partially observable rewards by proposing the Monitored MDP (Mon-MDP). In Mon-MDPs, the observability of the reward is dictated by a "Markovian entity" (the monitor). Thus, actions can have either immediate or long-term effects on the reward observability. For example, rewards may become observable only when certain conditions are met — such as when the agent presses a button in the environment, carries a special item, or operates in areas equipped with instrumentation. The control over observability of the reward opens avenues for model-based

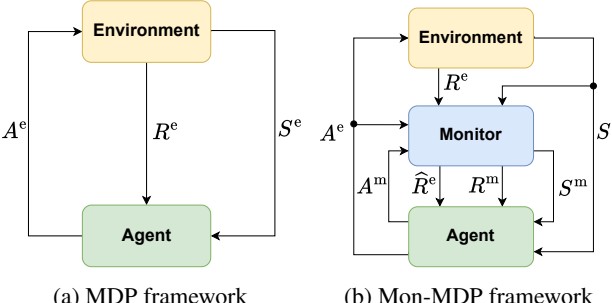

(a) MDP framework          (b) Mon-MDP framework

Figure 1: In MDPs (left) the agent interacts only with the environment and observes rewards at all times. In Mon-MDPs (right), the agent also interacts with the monitor, which dictates what rewards the agent observes. For clarity, we have omitted the dependence on time from the notation.

methods that attempt to model the process governing reward observability in order to *plan which rewards to observe, or which states to visit where rewards are more likely to be observed*. In the next sections, we 1) revisit Mon-MDPs as an extension of MDPs, 2) define minimax-optimality when some rewards may *never be observable*, and 3) highlight how our algorithm addresses existing limitations.

### 2.1. Monitored Markov Decision Processes

MDPs are represented by the tuple $\langle \mathcal{S}, \mathcal{A}, r, p, \gamma \rangle$, where $\mathcal{S}$ is the finite state space, $\mathcal{A}$ is the finite action space, $r : \mathcal{S} \times \mathcal{A} \to [r^{\text{e}}_{\min}, r^{\text{e}}_{\max}]$ is the mean reward function, $p : \mathcal{S} \times \mathcal{A} \to \Delta(\mathcal{S})$[2] is the Markovian transition dynamics, and $\gamma \in [0, 1)$ is the discount factor describing the trade-off between immediate and future gains. Mon-MDPs extend MDPs by introducing the *monitor*, another entity that the agent interacts with and is also governed by Markovian transition dynamics. Intuitively, Mon-MDPs incorporate two MDPs — one for the environment and one for the monitor — and we differentiate quantities associated with each using superscripts "e" and "m", respectively.

In Mon-MDPs, the state and the action spaces comprise the environment and the monitor spaces, i.e., $\mathcal{S} \coloneqq \mathcal{S}^{\text{e}} \times \mathcal{S}^{\text{m}}$ and $\mathcal{A} \coloneqq \mathcal{A}^{\text{e}} \times \mathcal{A}^{\text{m}}$. At every timestep, the agent observes the state of both the environment and the monitor, and performs a joint action. The monitor also has Markovian dynamics, i.e., $p^{\text{m}} : \mathcal{S}^{\text{m}} \times \mathcal{A}^{\text{m}} \times \mathcal{S}^{\text{e}} \times \mathcal{A}^{\text{e}} \to \Delta(\mathcal{S}^{\text{m}})$, and the joint transition dynamics is denoted by $p \coloneqq p^{\text{e}} \otimes p^{\text{m}} : \mathcal{S} \times \mathcal{A} \to \Delta(\mathcal{S})$. Note that the *monitor transition dynamics* depend on the *environment state and action*, highlighting the interplay between the monitor and the environment.

Mon-MDPs also have two mean rewards, $r \coloneqq (r^{\text{e}}, r^{\text{m}})$, where $r^{\text{m}} : \mathcal{S}^{\text{m}} \times \mathcal{A}^{\text{m}} \to [r^{\text{m}}_{\min}, r^{\text{m}}_{\max}]$ is also bounded (To reduce the clutter only in the theoretical part of this work, assume $r^{\text{e}}_{\min} \coloneqq -r^{\text{e}}_{\max} = -1$ and $r^{\text{m}}_{\min} \coloneqq -r^{\text{m}}_{\max} =$

---

[1]We denote random variables with capital letters.

[2]$\Delta(\mathcal{X})$ denotes the set of distributions over the finite set $\mathcal{X}$.

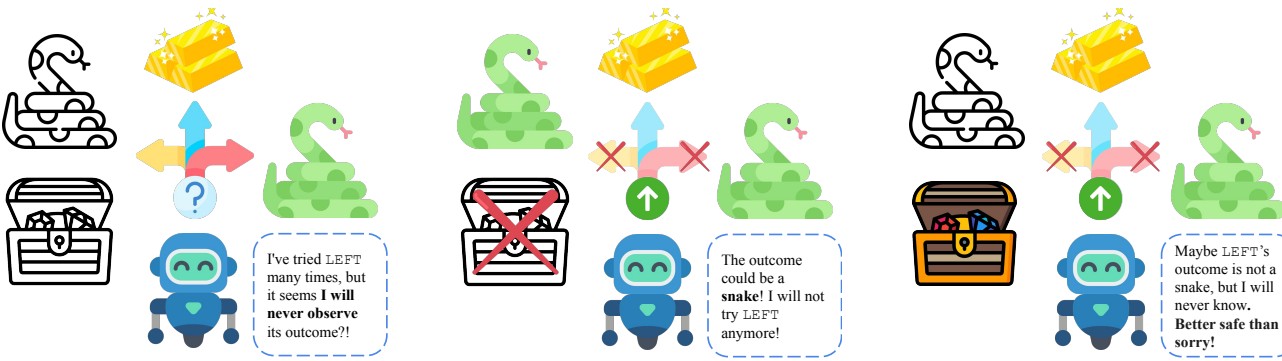

(a) LEFT's outcome cannot be observed and will always be unknown.

(b) A pessimistic agent assumes the worst (snake) for LEFT.

(c) The agent would be pessimistic even if LEFT's true outcome is not the snake.

Figure 2: **Example of a pessimistic agent in Mon-MDPs.** (a) The agent has to choose between LEFT, UP, and RIGHT. RIGHT leads to a snake, UP to gold bars, and LEFT to either a snake or a treasure chest (more valuable than gold bars), but the agent *can never observe the result* of executing LEFT. (b) After sufficient attempts[5], the agent excludes LEFT because its outcome is unknown and the agent assumes the worst. (c) LEFT is ruled out, even though it could actually yield the treasure chest. However, since this cannot be known, acting pessimistically complies with minimax-optimality in Equation (3). In the end, the agent explores new actions but it is also wary because some actions may never yield a reward. Thus, after enough exploration, it assumes the worst if the action's outcome is still unknown.

−1). However, unlike classical MDPs, *the environment rewards are not directly observable*. Instead, at a timestep $t$ in place of the (possibly stochastic) environment reward $R_{t+1}^e \in \mathbb{R}$, the agent observes the *proxy reward* $\widehat{R}_{t+1}^e \sim f^m\left(R_{t+1}^e, S_t^m, A_t^m\right)$, where $f^m : \mathbb{R} \times \mathcal{S}^m \times \mathcal{A}^m \to \mathbb{R} \cup \{\bot\}$ is the monitor function and $\bot$ denotes an "unobserved reward", i.e., the agent does not receive any numerical reward[3]. Using the above notation, a Mon-MDP $M$ can be compactly denoted by the tuple $M = \langle \mathcal{S}, \mathcal{A}, r, p, f^m, \gamma \rangle$. Figure 1 illustrates the agent-environment-monitor interaction.

In Mon-MDPs, at timestep $t$ the agent executes the joint action $A_t := (A_t^e, A_t^m)$ at the joint state $S_t := (S_t^e, S_t^m)$. In turn, the environment and monitor states change and produce a joint reward $(R_{t+1}^e, R_{t+1}^m)$, but the agent observes $\left(\widehat{R}_{t+1}^e, R_{t+1}^m\right)$. The agent's goal is to learn a policy $\pi : \mathcal{S} \to \Delta(\mathcal{A})$ selecting joint actions to maximize the expected[4] discounted return $\mathbb{E}\left[\sum_{t=0}^\infty \gamma^t \left(R_{t+1}^e + R_{t+1}^m\right)\right]$, *even though the agent observes $\widehat{R}_{t+1}^e$ instead of $R_{t+1}^e$*. This is the crucial difference between MDPs and Mon-MDPs: the immediate environment reward $R_{t+1}^e$ *is always generated by the environment*, i.e., desired behavior is well-defined as the reward is sufficient to describe the agent's task (Bowling et al., 2023). Yet, the monitor may "hide it" from the agent, possibly even yielding "unobservable reward" $\widehat{R}_{t+1}^e = \bot$

at all times for some state-actions. For example, consider a task where a human supervisor (the monitor) gives the reward: if the supervisor leaves, the agent will not observe any reward anymore; yet, the task has not changed, i.e., the human — *if present* — would still give the same rewards.

## 2.2. Learning Objective in Mon-MDPs

In this section, we define the state-values and action-values for Mon-MDPs. Further, we define the learning objective as finding a minimax-optimal policy. Let $\mathbb{P}$ and $\mathbb{E}$ be the probability measure and expectations we get when a policy $\pi$ is run in $M$ starting from an initial state. Similar to MDPs, the state-value $V_M^\pi$ and action-value $Q_M^\pi$ denote the expected sum of discounted rewards, with an optimal policy $\pi^*$ maximizing them:

$$V_M^\pi(s) := \mathbb{E}\left[\sum_{k=t}^\infty \gamma^{k-t} R_{k+1} \middle| S_t = s\right],$$

$$Q_M^\pi(s, a) := \mathbb{E}\left[\sum_{k=t}^\infty \gamma^{k-t} R_{k+1} \middle| S_t = s, A_t = a\right],$$

$$\underbrace{V_M^{\pi^*}}_{=:V_M^*} \in \sup_{\pi \in \Pi} V_M^\pi(s), \qquad \forall s \in \mathcal{S}, \qquad (1)$$

where $R_{k+1} := R_{k+1}^e + R_{k+1}^m$ is the immediate joint reward at timestep $k$ and $\Pi$ is the set of policies in $M$. We stress once more that the agent cannot observe the immediate environment reward $R_{k+1}^e$ directly and observes the

---

[3]Note that $f^m$ could return any arbitrary real number, unrelated to the environment reward. To rule out pathological cases (e.g., the monitor function always returns 0), $f^m$ is assumed to be truthful (Parisi et al., 2024b), i.e., the monitor either reveals the true environment reward ($\widehat{R}_{t+1}^e = R_{t+1}^e$) or hides it ($\widehat{R}_{t+1}^e = \bot$).

[4]Section 2.2 defines the reference measure for the expectation.

[5]We define "sufficient" in Theorem 3.1. Intuitively, the more confident the agent wants to be, the more it should try the action.

immediate proxy reward $\widehat{R}^{\mathrm{e}}_{k+1}$ for all $k \geq 0$, even though the environment still assigns rewards to the agent's actions.

Parisi et al. (2024a) showed asymptotic convergence to an optimal policy with an *ergodic* monitor function $f^{\mathrm{m}}$, i.e., for all environment state-action pairs $(s^{\mathrm{e}}, a^{\mathrm{e}})$, there exists at least a monitor state-action pair $(s^{\mathrm{m}}, a^{\mathrm{m}})$ such that the proxy reward is observed infinitely-often given infinite exploration. Formally, let $s := (s^{\mathrm{e}}, s^{\mathrm{m}})$ and $a := (a^{\mathrm{e}}, a^{\mathrm{m}})$, then

$$\mathbb{P}\left(\limsup_{t \to \infty} \widehat{R}^{\mathrm{e}}_{t+1} \neq \perp \,\middle|\, S_t = s, A_t = a\right) > 0.$$

Intuitively, this means the agent will always be able to observe every environment reward (infinitely many times, given infinite exploration). However, if even one environment reward is *never* observable, the Mon-MDP is *unsolvable*, and convergence to an optimal policy cannot be guaranteed. Essentially, if the agent can never know that a specific state-action yields the highest (or lowest) environment reward, then it can never learn to visit (or avoid) it. Nonetheless, we argue that assuming every environment reward is observable (sooner or later) is a very stringent condition, not suitable for real-world tasks — reward instrumentation may have limited coverage, human supervisors may never be available in the evening, or training before deployment may not guarantee full state coverage.

To define the learning objective even in unsolvable Mon-MDPs, we follow Parisi et al. (2024b, Appendix B.3). First, let $[M]_{\mathbb{I}}$ be the set of all Mon-MDPs the agent cannot distinguish based on the reward observability in $M$. That is, all Mon-MDPs with identical state and action spaces, transition dynamics and monitor function to $M$, but differing in their environment mean reward $r^{\mathrm{e}}$ (for the full definition see Appendix H). If $M$ is solvable, all rewards can be observed and $[M]_{\mathbb{I}} = \{M\}$, i.e., the set of indistinguishable Mon-MDPs to the agent is a singleton. Otherwise, from the agent's perspective, there are possibly infinitely many Mon-MDPs in $[M]_{\mathbb{I}}$ because the underlying never-observable rewards' mean could be any real value within their bounded range. Second, let $M_{\downarrow}$ be the worst-case Mon-MDP, i.e., the one whose all never-observable rewards are $r^{\mathrm{e}}_{\min}$:

$$M_{\downarrow} \in \operatorname*{argmin}_{M' \in [M]_{\mathbb{I}}} r^{\mathrm{e}}_{M'}(s^{\mathrm{e}}, a^{\mathrm{e}}), \quad \forall (s^{\mathrm{e}}, a^{\mathrm{e}}) \in \mathcal{S}^{\mathrm{e}} \times \mathcal{A}^{\mathrm{e}}, \quad (2)$$

where $r^{\mathrm{e}}_{M'}$ is the environment mean reward in $M'$. In words, $M_{\downarrow}$ is a Mon-MDP whose mean environment reward is minimized over all Mon-MDPs indistinguishable from $M$. Then, we define the *minimax-optimal policy* of $M$ as the optimal policy of the worst-case Mon-MDP, i.e.,

$$\underbrace{V^{\pi^*}_{M_{\downarrow}}}_{=:V^*_{\downarrow}} \in \sup_{\pi \in \Pi} V^{\pi}_{M_{\downarrow}}(s), \qquad \forall s \in \mathcal{S}, \qquad (3)$$

where $\Pi$ is the set of all policies in $M_{\downarrow}$. As noted earlier, if $M$ is solvable then $[M]_{\mathbb{I}} = \{M\}$, and Equation (3) is equivalent to Equation (1). Therefore, the minimax-optimal policy is simply the optimal policy. Figure 2 shows an unsolvable Mon-MDP and the minimax-optimal policy.

# 3. Monitored Model-Based Interval Estimation

We propose a novel model-based algorithm to exploit the structure of Mon-MDPs, show how to apply it on solvable and unsolvable Mon-MDPs, and provide sample complexity bounds. As our algorithm builds upon MBIE and MBIE-EB (Strehl & Littman, 2008), we first briefly review both.

## 3.1. MBIE and MBIE-EB

MBIE is an algorithm for learning an optimal policy in MDPs with polynomial sample complexity. MBIE maintains confidence intervals on all unknown quantities (i.e., mean rewards and transition dynamics) and solves the set of corresponding MDPs to produce an optimistic value function. Greedy actions with respect to this value function direct the agent toward insufficiently visited state-action pairs to be certain whether they are part of the optimal policy or not. MBIE-EB is a simpler variant that constructs a single confidence interval around the action-values (instead of building confidence intervals around the mean rewards and transitions separately) with an exploration bonus to be optimistic with respect to the uncertain quantities.

Let $\bar{R}$ and $\bar{P}$ be maximum likelihood estimates of the MDP's unknown mean rewards and transition dynamics based on the agent's experience, and let $N(s, a)$ count the number of times action $a$ has been taken in state $s$. MBIE-EB constructs an optimistic MDP[6],

$$\widetilde{R}(s, a) = \bar{R}(s, a) + \underbrace{\frac{\beta}{\sqrt{N(s, a)}}}_{\text{bonus for } r,\, p}, \qquad \widetilde{P} = \bar{P}, \quad (4)$$

where $\beta$ is a parameter chosen to be sufficiently large. It solves the MDP (using value iteration) to find $\widetilde{Q}$, the optimal action-value under the model, and acts greedily with respect to this function to gather more data to update its model.

## 3.2. Monitored MBIE-EB

Monitored MBIE-EB can be considered an extension of MBIE-EB to the Mon-MDPs with three key innovations. First, we adapt MBIE-EB to model each of the vital unknown components of the Mon-MDP (mean rewards and transition dynamics of both the environment and the monitor), each with their own exploration bonuses. Second, observing that the optimism for unobservable environment state-action pairs in an unsolvable Mon-MDP will never

---

[6]In this work, the denominator of bonuses starts at one; if zero, optimistic initialization is used unless otherwise stated.

vanish — the agent will try forever to observe rewards that are actually unobservable — we further adapt the algorithm to make worst-case assumptions on all unobserved environment rewards. Unfortunately, this creates an additional problem: environment state-action pairs whose rewards are hard to observe may never be sufficiently tried because they are dissuaded by the pessimistic model. Third, to balance the optimism-driven exploration and the pessimism induced by the worst-case assumption, Monitored MBIE-EB interleaves a second MBIE-EB instance that ensures unobserved environment state-actions are sufficiently explored.

**First Innovation: Extend MBIE-EB to Mon-MDPs.** Let $\bar{R}^{\text{e}}, \bar{R}^{\text{m}}, \bar{P}$ be maximum likelihood estimates of the environment mean reward, monitor mean reward, and the joint transition dynamics, respectively, all based on the agent's experience. Let $N(s^{\text{m}}, a^{\text{m}})$ count the number of times action $a^{\text{m}}$ has been taken in $s^{\text{m}}$, $N(s, a)$ count the same joint state-action pairs, and $N(s^{\text{e}}, a^{\text{e}})$ count environment state-action pairs, *but only if the environment reward was observed.* We then construct the following optimistic MDP using reward bonuses for the unknown estimated quantities,

$$\widetilde{R}_{\text{basic}}(s, a) = \bar{R}^{\text{e}}(s^{\text{e}}, a^{\text{e}}) + \overbrace{\frac{\beta^{\text{e}}}{\sqrt{N(s^{\text{e}}, a^{\text{e}})}}}^{\text{bonus for } r^{\text{e}}} +$$
$$\bar{R}^{\text{m}}(s^{\text{m}}, a^{\text{m}}) + \underbrace{\frac{\beta^{\text{m}}}{\sqrt{N(s^{\text{m}}, a^{\text{m}})}}}_{\text{bonus for } r^{\text{m}}} + \underbrace{\frac{\beta}{\sqrt{N(s, a)}}}_{\text{bonus for } p},$$
$$\widetilde{P} = \bar{P}, \tag{5}$$

where $\beta$, $\beta^{\text{e}}$, $\beta^{\text{m}}$ are hyperparameters for the confidence level of our optimistic model. As with MBIE-EB, this optimistic model is solved to find $\widetilde{Q}$ and actions are selected greedily. For solvable Mon-MDPs, MBIE-EB's theoretical results apply directly to the joint Mon-MDP, yielding a sample complexity bound. But, this algorithm fails to make any guarantees for unsolvable Mon-MDPs, where some environment rewards are never-observable. In such situations, $N(s^{\text{e}}, a^{\text{e}})$ never grows for some state-actions, thus optimism will direct the agent to seek out these state-actions, for which it can never reduce its uncertainty.

**Second Innovation: Pessimism Instead of Optimism.** We fix this excessive optimism in Equation (5) by creating a new reward model that is pessimistic, rather than optimistic, about unobserved environment state-action rewards:

$$\widetilde{R}_{\text{opt}}(s, a) = \begin{cases} \widetilde{R}_{\text{basic}}(s, a), & \text{if } N(s^{\text{e}}, a^{\text{e}}) > 0, \\ \widetilde{R}_{\text{min}}(s, a), & \text{otherwise,} \end{cases} \tag{6}$$

where $\widetilde{R}_{\text{min}}(s, a) =$

$$r_{\text{min}}^{\text{e}} + \bar{R}^{\text{m}}(s^{\text{m}}, a^{\text{m}}) + \frac{\beta^{\text{m}}}{\sqrt{N(s^{\text{m}}, a^{\text{m}})}} + \frac{\beta}{\sqrt{N(s, a)}}. \tag{7}$$

We call an episode where we take greedy actions according to $\widetilde{Q}_{\text{opt}}$ an *optimize* episode, as this ideally produces a minimax-optimal policy for all Mon-MDPs. The reader may have already realized this pessimism will introduce a new problem — dissuading the agent from exploring to observe previously unobserved environment rewards. Instead, we aim to observe all rewards but not too frequently that we prevent the agent from following the minimax-optimal policy when the Mon-MDP is actually unsolvable.

**Third Innovation: Explore to Observe Rewards.** We fix this now excessive pessimism by introducing a separate MBIE-EB-guided exploration aimed at discovering previously unobserved environment rewards. The following reward model does exactly that. Let $\mathbb{1}_{\{N(s^{\text{e}}, a^{\text{e}})=0\}}$ be the indicator function returning one if $N(s^{\text{e}}, a^{\text{e}})$ is equal to zero and returns zero otherwise, then $\widetilde{R}_{\text{obs}}(s, a) =$

$$\text{KL-UCB}\left(0, N(s, a)\right) \mathbb{1}_{\{N(s^{\text{e}}, a^{\text{e}})=0\}} + \frac{\beta^{\text{obs}}}{\sqrt{N(s, a)}}, \tag{8}$$

Therefore, the KL-UCB (Garivier & Cappé, 2011; Maillard et al., 2011) term is only included for environment state-action pairs whose rewards have not been observed. Let $d$ be the relative entropy between two Bernoulli distributions and $\beta^{\text{KL-UCB}}$ be a positive constant, then KL-UCB$(\bar{\mu}, n) =$

$$\max_{\mu} \left\{ \mu \in [0, 1] : d(\bar{\mu}, \mu) \leq \frac{\beta^{\text{KL-UCB}}}{n} \right\} \Bigg|_{\bar{\mu}=0} =$$
$$\max_{\mu} \left\{ \mu \in [0, 1] : \ln\left(\frac{1}{1-\mu}\right) \leq \frac{\beta^{\text{KL-UCB}}}{n} \right\}.$$

We are using KL-UCB to estimate an upper-confidence bound (with parameter $\beta^{\text{KL-UCB}}$) on the probability of observing the environment reward from joint state-action $(s, a)$ given that we have tried it $N(s, a)$ times already and have not succeeded to observe the reward (zero as the first argument of KL-UCB in $\widetilde{R}_{\text{obs}}$ indicates this lack of success). As we are constructing a upper confidence bound on a Bernoulli random variable (whether the reward is observed or not), KL-UCB is ideally suited and provides tight bounds. The result is an optimistic model for an MDP (built based on the joint Mon-MDP, where $\widetilde{R}_{\text{obs}}$ is the mean reward) that rewards the agent for observing previously unobserved environment rewards. An episode where we take greedy actions with respect to $\widetilde{Q}_{\text{obs}}$ we call an *observe* episode. If we have enough *observe* episodes, we can guarantee that all observable environment rewards are observed with high probability. If we have enough *optimize* episodes we can learn and follow the minimax-optimal policy. We balance between the two by switching the model we optimize according to

$$\widetilde{R}(s, a) = \begin{cases} \widetilde{R}_{\text{obs}}(s, a), & \text{if } \kappa \leq \kappa^*(k); \\ \widetilde{R}_{\text{opt}}(s, a), & \text{otherwise;} \end{cases} \tag{9}$$

where $\kappa^*$ is a sublinear function returning how many episodes of the $k$ total episodes should have been used to *observe* and $\kappa$ is the number of episodes that have been used to

*observe*. The choice of $\kappa^*$ is a hyperparameter. Monitored MBIE-EB then constructs the $\left\{ \widetilde{R}(s,a), \widetilde{P}(s,a) \right\}$ model and selects greedy actions with the respect to its optimal action-value $\widetilde{Q}$. The choice to hold the policy fixed throughout the course of an episode is a matter of simplicity, giving easier analysis that *observe* episodes will observe environment rewards, as well as computational convenience.

### 3.3. Theoretical Guarantees

One measure of RL algorithms' efficiency is their finite-time sample complexity, i.e., the number of timesteps (decision) required for the algorithm to find a near-desired policy with high probability (Kakade, 2003; Strehl et al., 2009; Lattimore & Hutter, 2012). Monitored MBIE-EB has a polynomial sample complexity (with respect to the relevant parameters) even in unsolvable Mon-MDPs. There exists parameters where the algorithm guarantees with high probability $(1-\delta)$ of being arbitrarily close $(\varepsilon)$ to a minimax-optimal policy for all but a finite number of timesteps, which is a polynomial of $\frac{1}{\varepsilon}$, $\frac{1}{\delta}$, and other quantities characterizing the Mon-MDP. Theorem 3.1 establishes the sample complexity of Monitored MBIE-EB, the first sample complexity bound for Mon-MDPs. It is a unification of necessary new propositions that extend the analysis of MBIE-EB (Strehl & Littman, 2008, Theorem 2) to Mon-MDPs.

**Theorem 3.1.** *For any $\varepsilon, \delta > 0$, and Mon-MDP $M$ where $\rho$ is the minimum non-zero probability of observing the environment reward in $M$ and $H$ is the maximum episode length, there exists constants $m_1$, $m_2$, and $m_3$, where*

$$m_1 = \widetilde{\mathcal{O}}\left( \frac{|\mathcal{S}|}{\varepsilon^2(1-\gamma)^4} \right),$$

$$m_2 = \widetilde{\mathcal{O}}\left( \frac{1}{\rho\varepsilon^2(1-\gamma)^4} \right),$$

$$m_3 = \widetilde{\mathcal{O}}\left( \frac{|\mathcal{S}|^2 |\mathcal{A}|}{\varepsilon^3(1-\gamma)^5} \right),$$

*such that Monitored MBIE-EB with parameters,*

$$\beta = \frac{2\gamma}{1-\gamma}\sqrt{2\ln\left(5|\mathcal{S}||\mathcal{A}|m_2/\delta\right)},$$

$$\beta^m = \sqrt{2\ln\left(5|\mathcal{S}||\mathcal{A}|m_2/\delta\right)},$$

$$\beta^e = \sqrt{2\ln\left(5|\mathcal{S}||\mathcal{A}|m_2/\delta\right)},$$

$$\beta^{obs} = (1-\gamma)^{-1}\sqrt{0.5\ln\left(10|\mathcal{S}||\mathcal{A}|m_1/3\delta\right)},$$

$$\beta^{KL\text{-}UCB} = \ln\left(10|\mathcal{S}||\mathcal{A}|m_1/3\delta\right),$$

$$\kappa^*(k) = m_3, \qquad \textit{(constant function)}$$

*provides the following bounds for $M$. Let $\pi_t$ denote Monitored MBIE-EB's policy and $S_t$ denote the state at time $t$. With probability at least $1-\delta$, $V_\downarrow^{\pi_t}(S_t) \geq V_\downarrow^* - \varepsilon$ for all but $\widetilde{\mathcal{O}}\left( \frac{|\mathcal{S}||\mathcal{A}|H}{\varepsilon^3(1-\gamma)^5\rho} \right)$ timesteps.*

The reader may refer to Appendix G for the proof.

An interesting addition to the bound over MBIE-EB bound for classical MDPs is the dependence on the $\rho$'s inverse, which bounds how difficult it is to observe the observable environment rewards. If a Mon-MDP does reveal all rewards (it is solvable) but only does so with infinitesimal probability, an algorithm must be suboptimal for many more timesteps. In Appendix I, in a simpler setting of a stochastic bandit problem with finitely-many arms and partially observable rewards by providing a lower bound, we show the dependence of Theorem 3.1's bound on $\rho^{-1}$ is essentially unimprovable.

### 3.4. Practical Implementation

The theoretically justified parameters for Monitored MBIE-EB present a couple of challenges in practice. First off, we rarely have a particular $\varepsilon$ and $\delta$ in mind, preferring algorithms that produce ever-improving approximations with ever-improving probability. Second, the bound, while polynomial in the relevant values, does not suggest practical parameters. The most problematic in this regard is the constant $\kappa^*$, which places all *observe* episodes at the start of training. Third, solving a model exactly and from scratch each episode to compute $\widetilde{Q}$ is computationally wasteful.

In practice, we slowly increase the confidence levels used in the bonuses over time. We follow the pattern of Lattimore & Szepesvári (2020), with the confidence growing slightly faster than logarithmically[7]. The scale parameters $\beta$, $\beta^m$, $\beta^e$, $\beta^{obs}$, $\beta^{KL\text{-}UCB}$ were tuned manually for each environment. For $\kappa^*$ we also grow it slowly over time allowing the agent to interleave *observe* and *optimize* episodes: $\kappa^*(k) = \log k$, where the log base was also tuned manually for each environment. Finally, rather than exactly solving the models every episode, we maintain two action-values: $\widetilde{Q}_{\text{obs}}$ and $\widetilde{Q}_{\text{opt}}$, both initialized optimistically. we do 50 steps of synchronous value iteration before every episode to improve $\widetilde{Q}_{\text{opt}}$, and before every *observe* episodes to improve $\widetilde{Q}_{\text{obs}}$.

## 4. Empirical Evaluation

This paper began by detailing limitations in prior work not leveraging the Mon-MDP structure, the possibility of a known monitor, nor dealing with unsolvable Mon-MDPs. This section breaks down this claim into four research questions (RQ) to investigate if Monitored MBIE-EB can: RQ1) Explore efficiently in hard-exploration tasks? RQ2) Act pessimistically when rewards are unobservable? RQ3) Identify

---

[7] We replace $\beta$ with $\beta\sqrt{g(\ln N(s))}$, where $g(x) = 1 + x\ln^2(x)$ and $N(s)$ counts the number of visits to state $s$. This choice of $g$ is required as rewards and state-values are unlikely to follow a Gaussian distribution, but being bounded allows us to assume they are sub-Gaussian. We similarly grow $\beta^e$, $\beta^m$, and $\beta^{obs}$, and replace $\beta^{KL\text{-}UCB}$ with $\beta^{KL\text{-}UCB}g(\ln N(s))$.

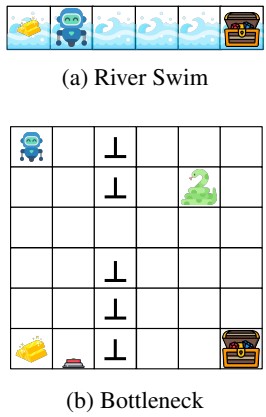

(a) River Swim

(b) Bottleneck

Figure 3: **Environments**

Monitored MBIE-EB    Directed-E²    Known Monitor    Minimax-Optimal

(a) River Swim     (b) Bottleneck (100%)     (c) Bottleneck (5%)     (d) Bottleneck (5%)

Figure 4: **Discounted return at test time**, averaged over 30 seeds (shaded areas denote 95% confidence intervals). Monitored MBIE-EB (in green) outperforms Directed-E² (in orange) and always converges to the minimax-optimal policy (the dashed black line). (c) and (d) both show results in the Bottleneck with the 5% Button Monitor, but with different axis ranges to highlight the improvement if Monitored MBIE-EB already knows details of the monitor (in purple).

and learn about difficult to observe rewards? RQ4) Take advantage of a known model of the monitor?

To directly address these questions, we first show results on two tasks with two monitors. Then, we show results on 48 benchmarks to strengthen our claims[8].

## 4.1. Environment and Monitor Description

*River Swim* (Figure 3a) is a well-known difficult exploration task with two actions. Moving LEFT always succeeds, but moving RIGHT may not — the river current may cause the agent to stay at the same location or even be pushed to the left. There is a goal state on the far right, where moving right yield a reward of 1. But, the leftmost tile yields 0.1 and it is easier to reach. Other states have zero rewards. Agents often struggle to find the optimal policy (always move RIGHT), and instead converge to always move LEFT. In our experiments, we pair River Swim with the *Full Monitor* where environment rewards are always freely observable, allowing us to focus on an algorithm's exploration ability.

*Bottleneck* (Figure 3b) has five deterministic actions: LEFT, UP, RIGHT, DOWN, STAY, which move the agent around the grid. Episodes end when the agent executes STAY in either the gold bars state (with a reward of 0.1) or in the treasure chest state (with a reward of 1). Reaching the snake state yields -10, and other states yield zero. However, states denoted by $\perp$ have *never-observable* rewards of -10, i.e., $R^e_{t+1} = -10$ but $\widehat{R}^e_{t+1} = \perp$ *at all times* $t$. In our experiments, we pair Bottleneck with the *Button Monitor*, where the monitor state can be ON or OFF (initialized at random) and is switched if the agent executes DOWN in the button state. When the monitor is ON, the agent receives $R^m_{t+1} = -0.2$ at every timestep $t$, but also observes the current environment reward (unless the agent is in a $\perp$ state). The minimax-optimal policy follows the shortest path to

the treasure chest, while avoiding the snake and $\perp$ states, and turning the monitor OFF if it was ON at the beginning of the episode. To evaluate Monitored MBIE-EB when observability is stochastic, we consider two Button Monitors: one where the monitor works as intended and rewards are observable with 100% probability if ON, and a second where the rewards are observable only with 5% probability if ON.

## 4.2. Results

We evaluate Monitored MBIE-EB compared to Directed Exploration-Exploitation (Directed-E²) (Parisi et al., 2024a), which is currently the most performant algorithm in Mon-MDPs. The discount factor is fixed to 0.99. Appendix E.1 contains the set of hyperparameters and Appendix B contains the evaluation details (e.g., episodes lengths, etc). Results in Figures 4 and 5 are at test time, i.e., when the agent follows the current greedy policy without exploring.

To answer RQ1, consider the results in Figure 4a. Monitored MBIE-EB significantly outperforms Directed-E². This task is difficult for any $\varepsilon$-greedy exploration strategy (such as the Directed-E²'s) and highlights the first innovation: taking a model-based approach in Mon-MDPs (i.e., extending MBIE-EB) leads to more efficient exploration.

To answer RQ2, consider Figure 4b. States marked with $\perp$ are never observable by the agent, regardless of the monitor state. Because the minimum mean reward in this task is $r_{\min} = -10$, the minimax-optimal policy is to avoid states marked by $\perp$ while reaching the goal state. Monitored MBIE-EB is able to find this minimax-optimal policy, whereas Directed-E² does not because it does not learn to avoid unobservable rewards[9]. This result highlights the

---

[8]Code: https://github.com/IRLL/Exploration-in-Mon-MDPs.

[9]Directed-E² describes initializing its reward model randomly, relying on the Mon-MDP being solvable, independent of the initialization. For unsolvable Mon-MDPs this is not true and Directed-E² depends significantly on initialization. In fact, while not noted

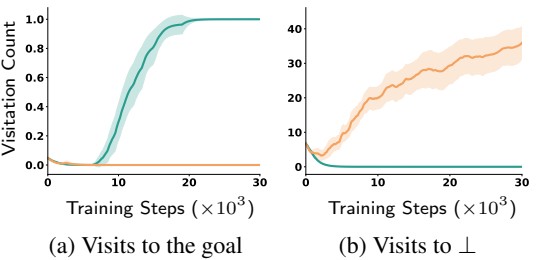

(a) Visits to the goal  (b) Visits to ⊥

Figure 5: **Visits to important states at test time in the Bottleneck with 5% Button Monitor**. Results are averaged over 30 trials, and shaded areas denote 95% confidence interval. Directed-$E^2$ fails to focus on the goal and instead keeps visiting ⊥ states, whereas Monitored MBIE-EB reduces its visitation frequency instead, ultimately visiting only the goal.

impact of the second innovation: unsolvable Mon-MDPS require pessimism when the reward cannot be observed.

To answer RQ3, consider Figure 4c, where the Button Monitor provides a reward only 5% of the time when ON (and 0% of the time when OFF). Despite how difficult it is to observe rewards, Monitored MBIE-EB is able to learn the minimax-optimal policy. This shows that Monitored MBIE-EB is still appropriately pessimistic, successfully avoiding ⊥ states and the snake, and reaching the goal state. Since rewards are only visible one out of twenty times (when the monitor is ON), learning is much slower than in Figure 4b, matching the appearance of $\rho^{-1}$ in Theorem 3.1's bound. This result also shows the impact of the third innovation: it is important to explore just enough to guarantee the agent will learn about observable rewards, but no more.

To answer RQ4, now consider the performance of Known Monitor in Figure 4d, showing the performance of Monitored MBIE-EB when provided the model of the Button Monitor 5%. Results show that its convergence speed increases significantly, as Monitored MBIE-EB takes (on average) 30% fewer steps to find the minimax-optimal policy. This feature of Monitored MBIE-EB is particularly important in settings where the agent has already learned about the monitor previously or the practitioner provides the agent with an accurate model of the monitor. The agent, then, needs only to learn about the environment, and does not need to explore the monitor component of the Mon-MDP.

To better understand the above results, Figure 5 shows how many times the agent visits the goal state and ⊥ states per testing episode. Both algorithms initially visit the goal state (Figure 5a) during random exploration (i.e., when executing the policy after 0 timesteps of training). Monitored MBIE-EB appropriately explores for some training episodes (recall that rewards are only observed in ON and even then only 5% of the time), and then learns to always go to the goal. Both also initially visit ⊥ states (Figure 5b). However, while Monitored MBIE-EB learns to be appropriately pessimistic over time and avoids them, Directed-$E^2$ never updates its (random) initial estimate of the value of ⊥ states and incorrectly believes they should continue to be visited. This also

by Parisi et al. (2024a), pessimistic initialization with Directed-$E^2$ results in asymptotic convergence for unsolvable Mon-MDPs.

explains why Directed-$E^2$ performs even worse in Figure 4c.

Finally, Figure 6 presents results comparing Monitored MBIE-EB across all of the domains and monitor benchmarks from Parisi et al. (2024a). In these 48 benchmarks, Monitored MBIE-EB significantly outperforms Directed-$E^2$ in all but five of them, where they perform similarly.

## 5. Discussion

There are a number of limitations to our approach suggesting directions for future improvements. First, Mon-MDPs contain an exploration-exploitation dilemma, but with an added twist — the agent needs to treat never observed rewards pessimistically to achieve a minimax-optimality; however, it should continue exploring those states to get more confident about their unobservability. Much like early algorithms for the exploration-exploitation dilemma in MDPs (Kearns & Singh, 2002), our approach separately optimizes a model for exploring and one for seeking minimax-optimality. A more elegant approach is to simultaneously optimize for both. Second, our approach uses explicit counts to drive its exploration, which limits it to enumerable Mon-MDPs. Adapting psuedocount-based methods (Bellemare et al., 2016; Martin et al., 2017; Tang et al., 2017; Machado et al., 2020) helps making Monitored MBIE-EB more applicable to large or continuous spaces. Finally, the decision of when to stop trying to observe rewards and instead optimize is essentially an optimal stopping time problem (Lattimore & Szepesvári, 2020), and there are considerable innovations that could improve the bounds along with empirical performance.

## 6. Conclusion

We introduced Monitored MBIE-EB for Mon-MDPs that addresses many of the previous work's shortcomings. It gives the first finite-time sample complexity for Mon-MDPs, while being applicable to both solvable and unsolvable Mon-MDPs, for which it is also the first. Furthermore, it both exploits the Mon-MDP's structure and leverage the monitor process's knowledge, if available. These features are not just theoretical, as we see these innovations resulting in empirical improvements in Mon-MDP benchmarks, outperforming the previous best learning algorithm.

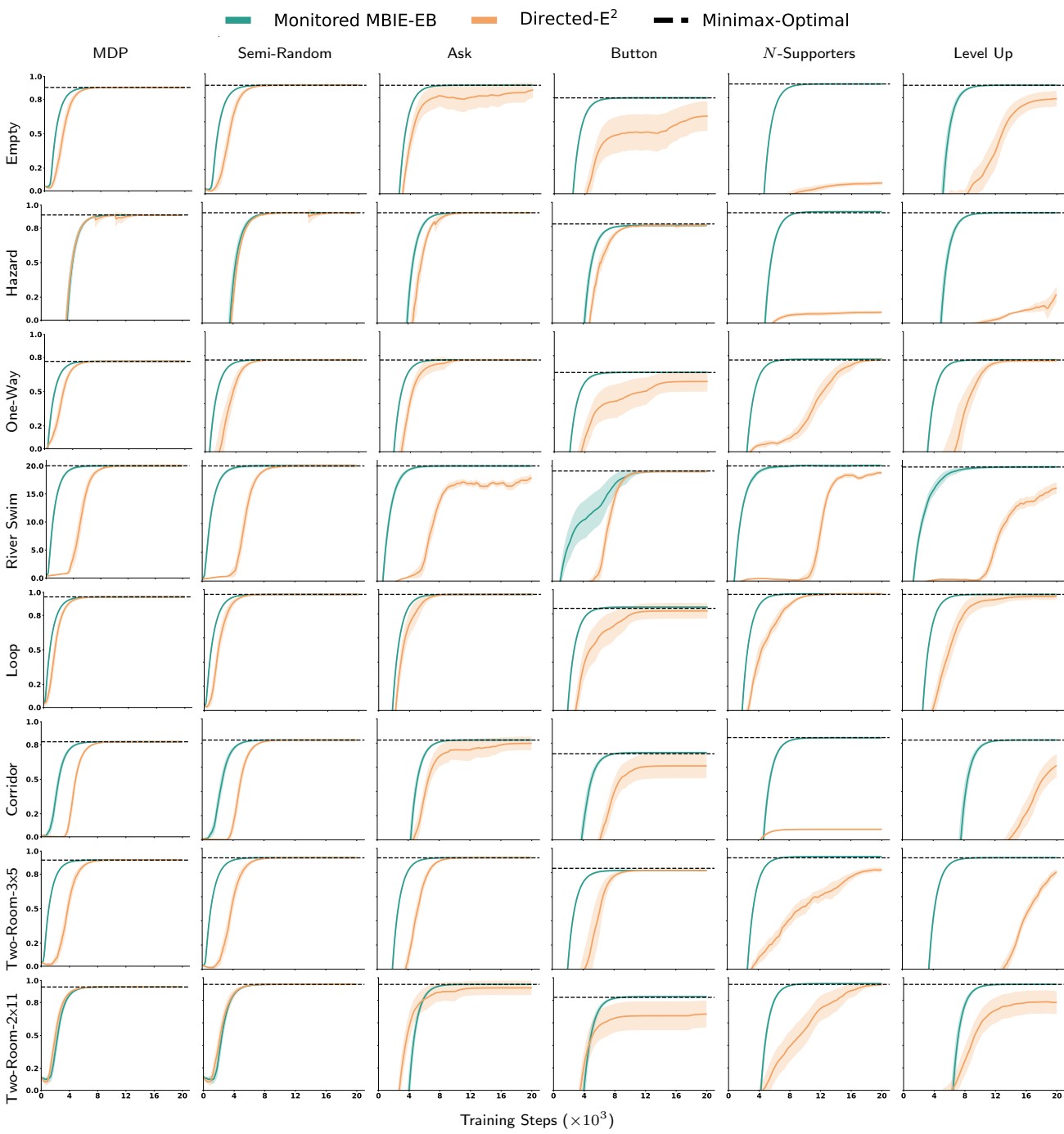

Figure 6: **Performance on 48 benchmark environments** from Parisi et al. (2024a). Monitored MBIE-EB outperforms Directed-E$^2$ in 43 of them and performs on par in the remainder five. Details of all 48 benchmarks are in Appendix B.

## Acknowledgments

The authors are grateful to Simone Parisi for helping in preparing this work, Antonie Bodley for enhancing the text's clarity, and the anonymous reviewers that provided valuable feedback. Part of this work has taken place in the Intelligent Robot Learning (IRL) Lab at the University of Alberta, which is supported in part by research grants from Alberta Innovates; Alberta Machine Intelligence Institute (Amii); a Canada CIFAR AI Chair, Amii; Digital Research Alliance of Canada; Mitacs; and the National Science and Engineering Research Council (NSERC).

## Impact Statement

This paper presents work whose goal is to advance the fundamental understanding of reinforcement learning. Our work is mostly theoretical and experiments are conducted on simple environments that do not involve human participants or concerning datasets, and it is not tied to specific real-world applications. We believe that our contribution has little to no potential for harmful impact.

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

In this appendix we provide the proof of the sample complexity bound of Monitored MBIE-EB, the description of how the set of indistinguishable Mon-MDP $[M]_{\mathbb{I}}$ for a truthful Mon-MDP is defined, the proof of the tightness of the Monitored MBIE-EB's sample complexity on $\rho^{-1}$, additional details about Monitored MBIE-EB and its pseudocode, the details of experiments (including description of the environments and monitors, details of the hyperparameters, an outline of Directed-E$^2$, and left-out implementation details), and ablations on the significant components of Monitored MBIE-EB.

# A. Table of Notation

| Symbol | Explanation |
|---|---|
| $\Delta(\mathcal{X})$ | $\Delta(\mathcal{X})$ denotes the set of distributions over the finite set $\mathcal{X}$ |
| $M$ | A Mon-MDP |
| $\mathbb{P}$ | The probability measure |
| $\mathbb{E}$ | The expectation with respect to $\mathbb{P}$ |
| $\mathcal{S}$ | State space (environment-only in MDPs, joint environment-monitor in Mon-MDPs) |
| $\mathcal{A}$ | Action space (environment-only in MDPs, joint environment-monitor in Mon-MDPs) |
| $r(s, a)$ | Mean reward of taking action $a$ at state $s$ |
| $p$ | Transition dynamics in MDPs / Joint transition dynamics in Mon-MDPs |
| $\bar{P}$ | Maximum likelihood estimate of $p$ |
| $\widetilde{P}$ | Optimistic variant of $p$ |
| $\gamma$ | Discount factor |
| $r^{\mathrm{e}}_{\max}$ | Maximum mean environment reward |
| $r^{\mathrm{e}}_{\min}$ | $-r^{\mathrm{e}}_{\max}$ |
| $\mathcal{S}^{\mathrm{e}}$ | Environment state space |
| $\mathcal{A}^{\mathrm{e}}$ | Environment action space |
| $R^{\mathrm{e}}_{t+1}$ | Immediate environment reward for timestep $t$ |
| $\widehat{R}^{\mathrm{e}}_{t+1}$ | Immediate environment proxy reward for timestep $t$ |
| $\bar{R}^{\mathrm{e}}$ | Maximum likelihood estimate of $r^{\mathrm{e}}$ |
| $\widetilde{R}^{\mathrm{e}}$ | Optimistic variant of $r^{\mathrm{e}}$ |
| $p^{\mathrm{e}}$ | Environment transition dynamics |
| $\mathcal{S}^{\mathrm{m}}$ | Monitor state space |
| $p^{\mathrm{m}}$ | Monitor transition dynamics |
| $f^{\mathrm{m}}$ | Monitor function |
| $r^{\mathrm{m}}_{\max}$ | Maximum expected monitor reward |
| $r^{\mathrm{m}}_{\min}$ | $-r^{\mathrm{m}}_{\max}$ |
| $N(s, a)$ | Number of times $a$ has been taken at $s$ |
| $N(s^{\mathrm{m}}, a^{\mathrm{m}})$ | Number of times $a^{\mathrm{m}}$ has been taken at $s^{\mathrm{m}}$ |
| $N(s^{\mathrm{e}}, a^{\mathrm{e}})$ | Number of times $a^{\mathrm{e}}$ has been taken at $s^{\mathrm{e}}$ and the environment reward observed |
| $\kappa$ | Number of observe episodes |
| $\kappa^*(k)$ | Desired number of observe episodes through episode $k$ |
| $\rho$ | The minimum non-zero probability of observing the environment reward |

# B. Methodological Details and Results

Throughout the paper, our baseline is Directed-E$^2$ as the precursor of our work. Parisi et al. (2024a) showed the superior Directed-E$^2$'s performance in Mon-MDPs against many well-established algorithms. Directed-E$^2$ uses two action-values, one as the ordinary action-values that uses a reward model in place of the environment reward to update task-respecting action-values (this sets the agent free from the partial observability of the environment reward once the reward model is learned). The Second action-value denoted by $\Psi$, dubbed as visitation-values, tries to maximize the successor representations. Directed-E$^2$ uses visitation-values to visit every joint state-action pairs and it tries to maintain the visitation of each pair in a comparable range. In the limit of infinite exploration, Directed-E$^2$ becomes greedy with respect to the task-respecting action-values for maximizing the expected sum of discounted rewards.

Evaluation is based on discounted test return, averaged over 30 seeds with 95% confidence intervals. Every 100 steps, learning is paused, the agent is tested for 100 episodes, and the mean return is recorded

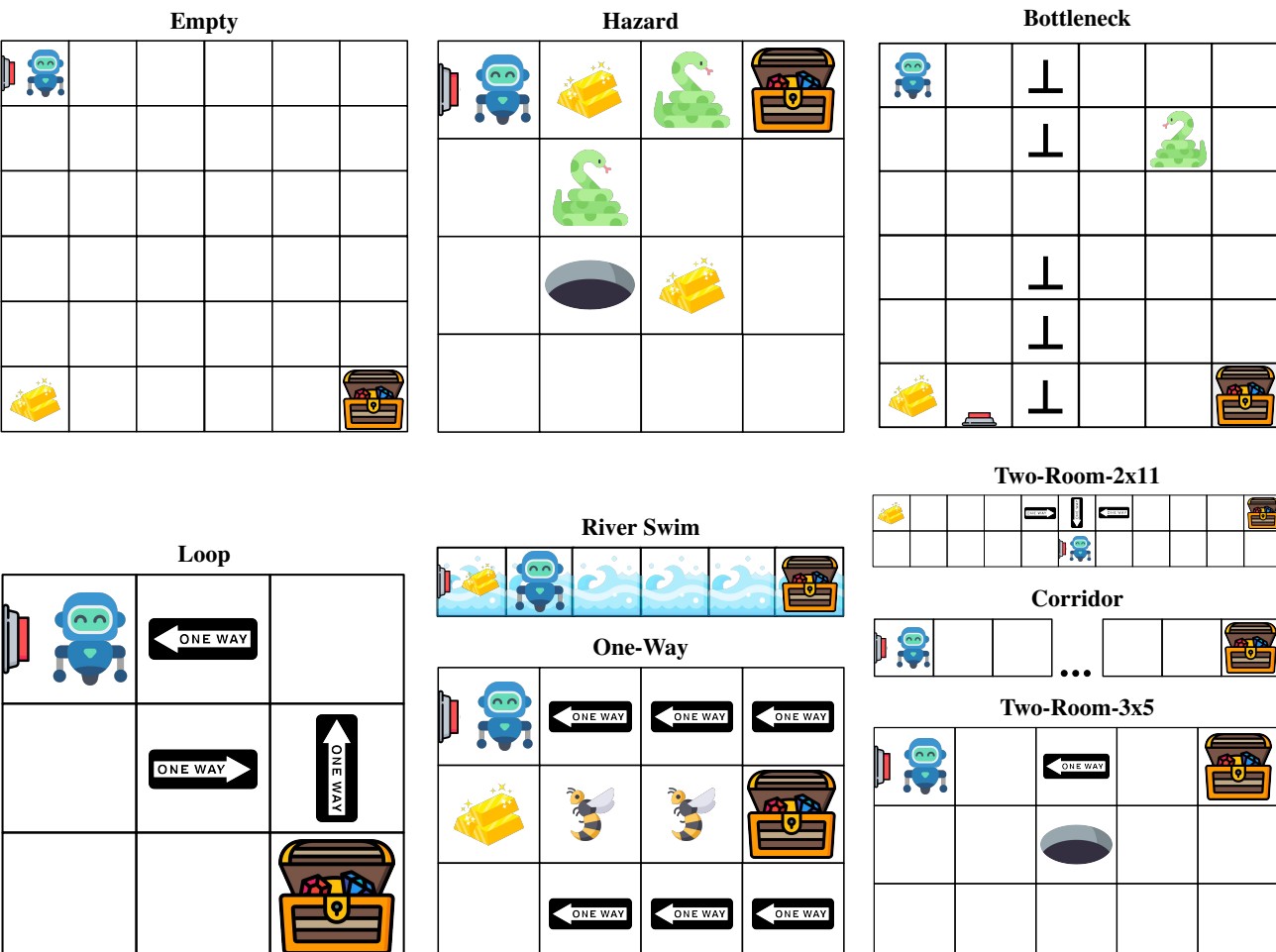

Figure 7: **Full set of environments**. Except Bottleneck, all environments are borrowed from Parisi et al. (2024a). Snakes and Wasps should be avoided. The goal is to STAY at the treasure chest. Gold bars are distractors. The agent gets stuck in the holes unless randomly gets pulled out. One-ways transition the agent in their own direction regardless of the action.

## B.1. Full Environments' Details

The environments comprise **Empty**, **Hazard**, **Bottleneck**, **Loop**, **River Swim**, **One-Way**, **Corridor**, **Two-Room-3x5** and **Two-Room-2x11**. They are shown in Figure 7. In all of them (except River Swim) the agent, represented by the robot, has 5 actions including 4 cardinal movement {LEFT, DOWN, RIGHT, UP} and an extra STAY action. The goal is getting to the treasure chest as fast as possible and STAYing there to get a reward of +1, which also terminates the episode. The agent should not STAY at gold bars as they are distractors because they would yield a reward of 0.1 and terminate the episode. The agent should avoid Snakes as any action leading to them yield a reward of -10. Cells with a one-way sign transition the agent only to their unique direction. If the agent stumbles on a hole, it would spend the whole episode in it, unless with 10% chance its action gets to be effective and the agent gets transitioned. When a button monitor is configured on top of the environment, the location of the button is figuratively indicated by a push button. The button is pushed if the agent bumps itself to it. The episode's time limit in River Swim, corridor and Two-Room-2x11 is 200 steps, and in other environments is 50 steps. In River Swim the agent has two actions L ≡ LEFT and R ≡ RIGHT, and there is no termination except the episode's time limit. In this environment gold bars have a reward of 0.01. Figure 8 shows the dynamics of River Swim.

## B.2. Full Monitors' Details

Monitors used in the experiments are: **Full (MDP)**, **Semi-Random**, **Full-Random**, **Ask**, **Button**, $N$-**Supporters**, $N$-**Experts** and **Level-Up**. For any of the monitors, except **Full-Monitor**, if a cell in the environment is marked with ⊥, then under no circumstances and at no timestep, the monitor would reveal the environment reward to agent for the action that led

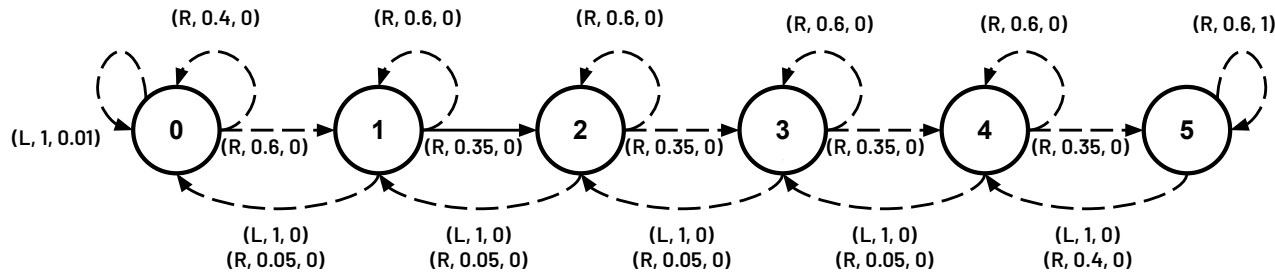

Figure 8: **Dynamics of River Swim**. Each tuple represents (action, transition probability, reward).

the agent to that cell. For *the rest* of the environment state-action pairs the behavior of monitors at timestep $t \geq 0$, by letting $X_t \sim \mathcal{U}[0, 1]$, where $\mathcal{U}$ is the uniform distribution and $\rho \in [0, 1]$, is as follows :

- **Full**. This corresponds to the MDP setting. Monitor shows the environment reward for all environment state-action pairs. State space and action state space of the monitor are singletons and the monitor reward is zero:

$$\mathcal{S}^m := \{\text{ON}\}, \qquad \mathcal{A}^m := \{\text{NO-OP}\}, \qquad S^m_{t+1} := \text{ON}, \qquad R^m_{t+1} := 0, \qquad \widehat{R}^e_{t+1} := R^e_{t+1}.$$

- **Semi-Random**. It is similar to Full-Monitor except that non-zero environment rewards get hidden with 50% chance:

$$\mathcal{S}^m := \{\text{ON}\}, \qquad \mathcal{A}^m := \{\text{NO-OP}\}, \qquad S^m_{t+1} := \text{ON}, \qquad R^m_{t+1} := 0. \qquad \widehat{R}^e_{t+1} := \begin{cases} R^e_{t+1}, & \textbf{if } R^e_{t+1} = 0; \\ R^e_{t+1}, & \textbf{else if } X_t \leq 0.5 \\ \bot, & \text{Otherwise.} \end{cases}$$

- **Full-Random**. It is similar to Semi-Random except *any* environment reward gets hidden with a predefined chance $1 - \rho$:

$$\mathcal{S}^m := \{\text{ON}\}, \qquad \mathcal{A}^m := \{\text{NO-OP}\}, \qquad S^m_{t+1} := \text{ON}, \qquad R^m_{t+1} := 0, \qquad \widehat{R}^e_{t+1} := \begin{cases} R^e_{t+1} & \textbf{if }, X_t \leq \rho; \\ \bot, & \text{Otherwise.} \end{cases}$$

- **Ask**. The monitor state space is a singleton but its action space has two elements: $\{\text{ASK}, \text{NO-OP}\}$. The agent gets to see the environment reward with probability $\rho$ if it ASKs. Upon ASKing the agent pays -0.2 as the monitor reward:

$$\mathcal{S}^m := \{\text{ON}\}, \qquad \mathcal{A}^m := \{\text{ASK, NO-OP}\}, \qquad S^m_{t+1} := \text{ON},$$

$$\widehat{R}^e_{t+1} := \begin{cases} R^e_{t+1}, & \textbf{if } X_t \leq \rho \textbf{ and } A^m_t = \text{ASK}; \\ \bot, & \text{Otherwise}; \end{cases} \qquad R^m_{t+1} := \begin{cases} -0.2, & \textbf{if } A^m_t = \text{ASK}; \\ 0, & \text{Otherwise.} \end{cases}$$

- **Button**. The state space is $\{\text{ON, OFF}\}$. The action space is a singleton. The agent sees the environment reward with probability $\rho$ as long as the monitor is ON, while paying -0.2. The state is flipped if the agent bumps itself to the button:

$$\mathcal{S}^m := \{\text{OFF, ON}\}, \qquad \mathcal{A}^m := \{\text{NO-OP}\}, \qquad \widehat{R}^e_{t+1} := \begin{cases} R^e_{t+1}, & \textbf{if } X_t \leq \rho \textbf{ and } S^m_t = \text{ON}; \\ \bot, & \text{Otherwise}; \end{cases}$$

$$S^m_{t+1} := \begin{cases} \text{ON}, & \textbf{if } S^m_t = \text{OFF and } S^e_t = \text{"BUTTON-CELL" and } A^e_t = \text{"BUMP-INTO-BUTTON"}; \\ \text{OFF}, & \textbf{if } S^m_t = \text{ON and } S^e_t = \text{"BUTTON-CELL" and } A^e_t = \text{"BUMP-INTO-BUTTON"}; \\ S^m_t, & \text{Otherwise}; \end{cases}$$

$$S^m_0 := \text{Random uniform from } \mathcal{S}^m, \qquad R^m_{t+1} := \begin{cases} -0.2, & \textbf{if } S^m_t = \text{ON}; \\ 0, & \text{Otherwise.} \end{cases}$$

- $N$-**Supporters**. The monitor state space comprises $N$ states that each represents the presence of a supporter. The action space also comprises $N$ actions. At each timestep one of the supporters is randomly present and if the agent could

choose the action that matches the index of the present supporter, then the agent gets to see the environment reward with probability $\rho$. Upon observing the environment reward agent pays a penalty of $-0.2$ as the monitor reward. However, if the agent chooses a wrong supporter, then it will be rewarded $0.001$ (as distraction) as the monitor reward:

$$\mathcal{S}^{\mathrm{m}} := \{0, \cdots, N-1\}, \qquad \mathcal{A}^{\mathrm{m}} := \{0, \cdots, N-1\}, \qquad S_{t+1}^{\mathrm{m}} := \text{Random uniform from } \mathcal{S}^{\mathrm{m}},$$

$$\widehat{R}_{t+1}^{\mathrm{e}} := \begin{cases} R_{t+1}^{\mathrm{e}}, & \textbf{if } X_t \le \rho \textbf{ and } S_t^{\mathrm{m}} = A_t^{\mathrm{m}}; \\ \perp, & \text{Otherwise}; \end{cases} \qquad R_{t+1}^{\mathrm{m}} := \begin{cases} -0.2, & S_t^{\mathrm{m}} = A_t^{\mathrm{m}}; \\ 0.001, & \text{Otherwise}. \end{cases}$$

Parisi et al. (2024a) considered this monitor as challenging, due to its bigger spaces than the rest of the monitors, for algorithms that use the successor representations for exploration. However, the encouraging nature of the monitor regarding the agent's mistakes makes the monitor easy for reward-respecting algorithms, e.g., Monitored MBIE-EB.

- $N$-**Experts**. Similar to $N$-Supporter the state space has $N$ states, each corresponding to the presence of one of the $N$ experts. However, getting experts' advice is costly, hence the action space has $N+1$ action where the last action corresponds to not pinging any experts and is cost-free. At each timestep, one of the experts is randomly present and if the agent selects the action that matches the present expert's index, the agent gets to see the environment reward with probability $\rho$. Upon observing the environment reward agent pays a penalty of $-0.2$ as the monitor reward. However, if the agent chooses a wrong expert it will be penalized by $-0.001$ as the monitor reward. Since the last action does not inquire any of the experts its monitor reward is $0$:

$$\mathcal{S}^{\mathrm{m}} := \{0, \cdots, N-1\}, \qquad \mathcal{A}^{\mathrm{m}} := \{0, \cdots, N\}, \qquad S_{t+1}^{\mathrm{m}} := \text{Random uniform from } \mathcal{S}^{\mathrm{m}},$$

$$\widehat{R}_{t+1}^{\mathrm{e}} := \begin{cases} R_{t+1}^{\mathrm{e}}, & \textbf{if } X_t \le \rho \textbf{ and } S_t^{\mathrm{m}} = A_t^{\mathrm{m}}; \\ \perp, & \text{Otherwise}; \end{cases} \qquad R_{t+1}^{\mathrm{m}} := \begin{cases} -0.2, & \textbf{if } S_t^{\mathrm{m}} = A_t^{\mathrm{m}}; \\ 0, & \textbf{else if } A_t^{\mathrm{m}} = N; \\ -0.001, & \text{Otherwise}. \end{cases}$$

- **Level-Up**. This monitor tries to test the agent's capabilities of performing deep exploration (Osband et al., 2019) in the monitor spaces. The state space has $N$ states corresponding to $N$ levels. The action space has $N+1$ actions. The initial state of the monitor is $0$ and if at each timestep the agent selects the action that matches the state of the monitor, the state increases by one. If the agent selects the wrong action the state is reset back to $0$. The agent only gets to observe the environment reward with probability $\rho$ if it gets the state of the monitor to the max level. The agent is penalized with $-0.2$ as the monitor reward every time it does not select the last action which does nothing and keeps the state as is:

$$\mathcal{S}^{\mathrm{m}} := \{0, \cdots, N-1\}, \qquad \mathcal{A}^{\mathrm{m}} := \{0, \cdots, N-1, \texttt{NO-OP}\}, \qquad \widehat{R}_{t+1}^{\mathrm{e}} := \begin{cases} R_{t+1}^{\mathrm{e}}, & \textbf{if } X_t \le \rho \textbf{ and } S_t^{\mathrm{m}} = N-1; \\ \perp, & \text{Otherwise}; \end{cases}$$

$$R_{t+1}^{\mathrm{m}} := \begin{cases} 0, & \textbf{if } A_t^{\mathrm{m}} = \texttt{NO-OP}; \\ -0.2, & \text{Otherwise}; \end{cases} \qquad S_{t+1}^{\mathrm{m}} := \begin{cases} S_t^{\mathrm{m}}, & \textbf{if } A_t^{\mathrm{m}} = \texttt{NO-OP}; \\ \max\{S_t^{\mathrm{m}}+1, N-1\}, & \textbf{else if } S_t^{\mathrm{m}} = A_t^{\mathrm{m}}; \\ 0, & \text{Otherwise}. \end{cases}$$

### B.3. When There Are Never-Observable Rewards

Mon-MDPs designed by Parisi et al. (2024a) do not have non-ergodic monitors, which give rise to unsolvable Mon-MDPs. Hence, we introduce **Bottleneck** to investigate the performance of Monitored MBIE-EB compared to Directed-$E^2$ in unsolvable Mon-MDPs. In Bottleneck the underlying reward of cells marked by $\perp$ is the same as the snake (-10). In these experiments, we use Full-Random monitor since it is more stochastic than Semi-Random to increase the difficulty. Results are shown in Figure 9. The location of the button is chosen to make agents perform deep exploration because changing the state of the button requires a long sequence of costly actions, which in turn yields the highest return. Therefore, the range of returns obtained with Button monitor is naturally lower than the rest of the Mon-MDPs.

As noted in Footnote 9, Directed-$E^2$'s performance in unsolvable Mon-MDPs depends critically on its reward model initialization. In order to see minimax-optimal performance from that algorithm, we need to initialize it pessimistically. As shown in Figures 4b and 4c, using the recommended random initialization saw essentially no learning in these domains. The algorithm would believe the never-observable rewards were at their initialized value, and so seek them out rather than treat their value pessimistically. Also, one of the weaknesses of Directed-$E^2$ is its explicit dependence on the size of the

state-action space during its exploration phase, i.e., Directed-$E^2$ tries to visit every joint state-action pair infinitely often without paying attention to their importance on maximizing the return. As a result, as the state-action space gets larger, the performance of Directed-$E^2$ deteriorates. To highlight the Directed-$E^2$'s weakness in larger spaces, we use $N$-Experts monitor as an extension of Ask monitor; Ask is the special case when $N = 1$. Figure 9 shows Directed-$E^2$'s performance is hindered when the agent faces $N$-Experts compared to Ask, while Monitored MBIE-EB suffers to a lesser degree.

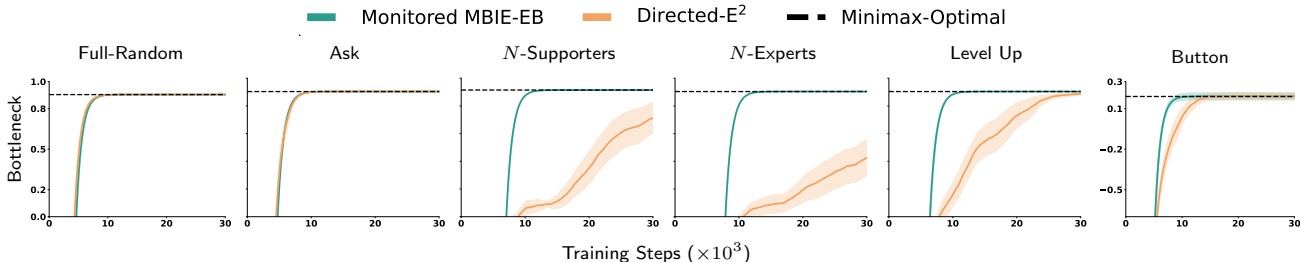

Figure 9: **Monitored MBIE-EB outperforms Directed-$E^2$ on Bottleneck with a non-ergodic monitor**. Even though Directed-$E^2$'s reward model was initialized pessimistically to achieve asymptotic minimax-optimality, its dependence on the state and action spaces' size makes it struggle more than Monitored MBIE-EB on $N$-Supporters, $N$-Experts and Level Up.

## B.4. When There Are Stochastically-Observable Rewards

In all the previous experiments, had agent done the action that would have revealed the environment reward, e.g., ASKing in Ask monitor, by paying the cost it would have observed the reward with 100% certainty. But, even if the probability is not 100% and yet bigger than zero, upon *enough* attempts to observe the reward and paying the cost, *even if a portion of the attemps are fruitless*, it is still possible to observe and learn the environment mean reward. In the Figure 10's experiments, in addition to have environment state-action pairs that their reward is permanently unobservable, we make the monitor stochastic for other pairs, i.e., even if the agent pays the cost, it would only observe the reward only with probability $\rho$. Figure 10 illustrates that albeit the challenge of having stochastically and also permanently unobservable rewards, Monitored MBIE-EB has not become prematurely pessimistic about the rewards that can be effectively observed, even when the probability goes down as low as 5%, and still Monitored MBIE-EB outperforms Directed-$E^2$.

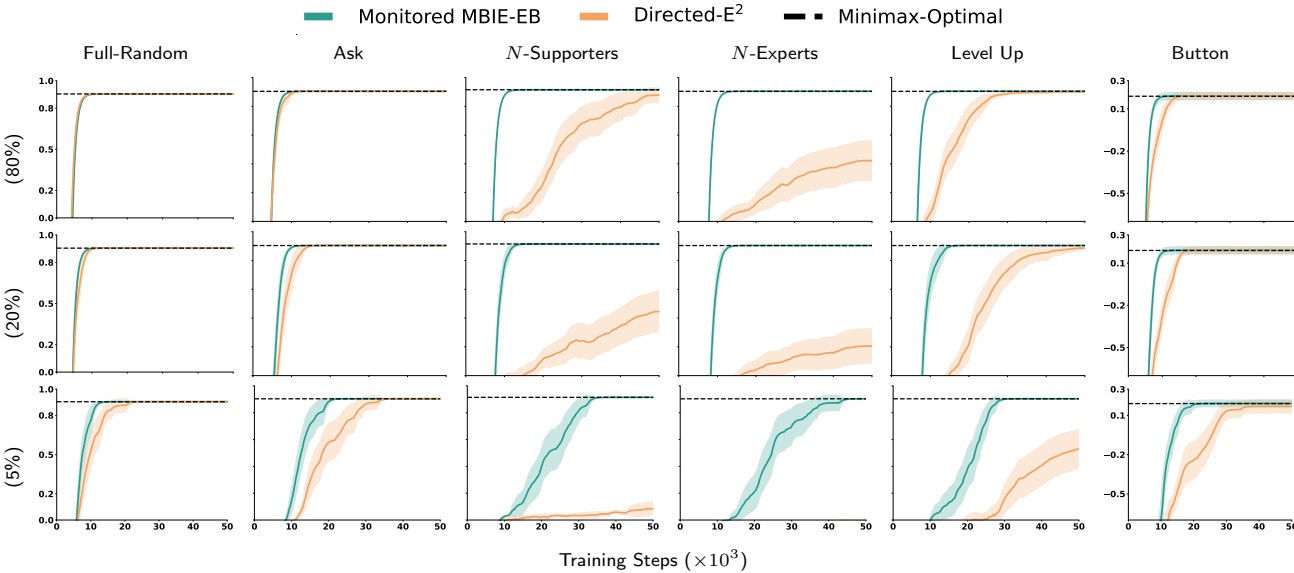

Figure 10: **Monitored MBIE-EB outperforms Directed-$E^2$ on Bottleneck even if environment rewards are stochastically observable**. The plots empirically show the effect of $\rho^{-1}$ in the Monitored MBIE-EB's sample complexity stated in Theorem 3.1. For a fixed environment and monitor, as the probability of observing the reward decreases, more samples are required to find a minimax-optimal policy. The plots also indicates that although the sample complexity of Directed-$E^2$ has never been given theoretically, but it must more severely depend on $\rho^{-1}$ than Monitored MBIE-EB's.

### B.5. When The Monitor is Known

In this section, we verify knowing the monitor speeds up the Monitored MBIE-EB's learning. We empirically demonstrate knowing the monitor is an advantage Monitored MBIE-EB can benefit from, while it is not readily possible in a model-free algorithm such as Directed-$E^2$. So far, we have shown the superior performance of Monitored MBIE-EB against Directed-$E^2$, now we investigate how much of the learning's difficulty in Mon-MDPs comes from the monitor being unknown. The unknown quantities of the monitor are $r^m$, $p^m$, and $f^m$, hence in the Figure 11's experiments we make all of them known to the agent in advance. The only remaining unknowns are $r^e$ and $p^e$. Hence, we replace Equation (8) with

$$\widetilde{R}_{\text{obs}}(s,a) = \mathbb{P}\left(\widehat{R}_{t+1} \neq \bot \,\Big|\, S_t = s, A_t = a\right) + \beta\sqrt{\frac{g(N_v(s^e))}{N_v(s^e, a^e)}},$$

where $N_v(s^e, a^e)$ counts the number of times $s^e, a^e$ has been visited, $N_v(s^e) = \sum_a N_v(s^e, a^e)$ and $g$ is defined in Footnote 7. The *intuition* behind the bonus $\beta\sqrt{\frac{g(N_v(s^e))}{N_v(s^e,a^e)}}$ is that $p = p^e \otimes p^m$ and we only need to account for the uncertainty stemming from not knowing $p^e$. Note since the monitor is known there is no need to use KL-UCB, as the agent already knows which environment rewards are observable (with what probability) and which ones are not. Similarly, we replace Equation (5) with

$$\widetilde{R}_{\text{basic}}(s,a) = \bar{R}^e(s^e, a^e) + \beta^e\sqrt{\frac{g(N(s^e))}{N(s^e, a^e)}} + \bar{R}^m(s^m, a^m) + \beta\sqrt{\frac{g(N_v(s^e))}{N_v(s^e, a^e)}},$$

where the bonus $\beta^e\sqrt{\frac{g(N(s^e))}{N(s^e,a^e)}}$ is due to the environment mean reward, and $\beta\sqrt{\frac{g(N_v(s^e))}{N_v(s^e,a^e)}}$ accounts for the fact that $\bar{p}^e$ only gets more accurate by visiting insufficiently visited environment state-actions. Figure 11 shows the prior knowledge of the monitor boosts the speed of Monitored MBIE-EB's learning and make it robust even in the low probability regimes.

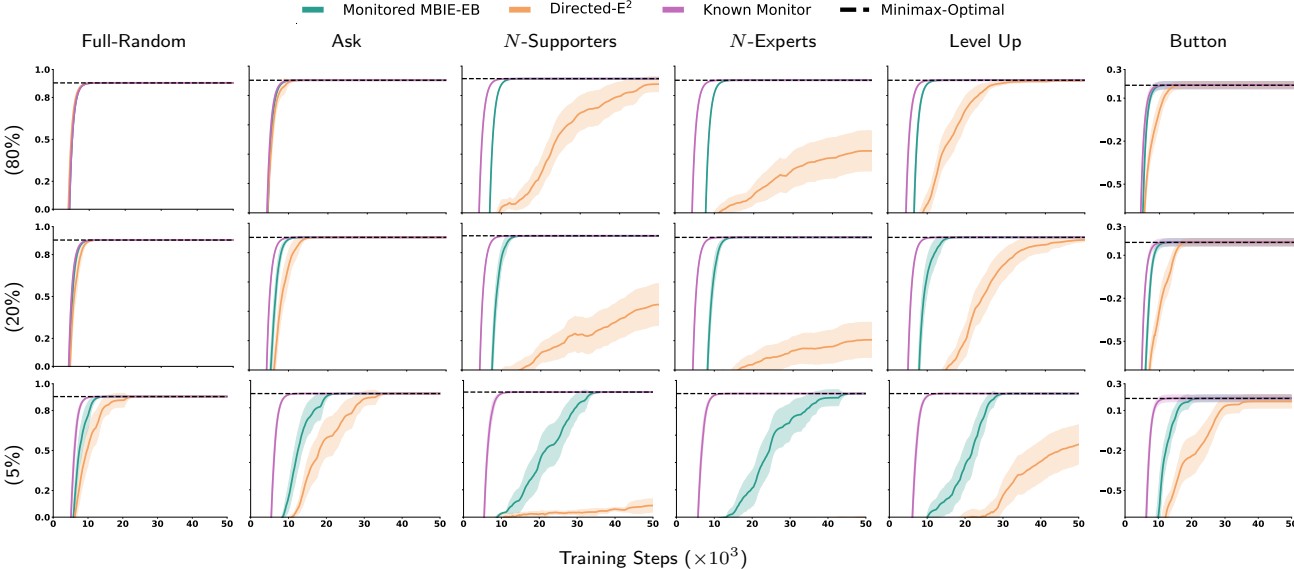

Figure 11: **Knowing the monitoring process considerably accelerates learning in Mon-MDPs**. The similar learning speed in Ask and $N$-Experts show that the knowledge of the monitor make Monitored MBIE-EB robust against the size of the monitor spaces. Also, the similarity of learning speed for a fixed environment and monitor across experiments with high and low observability chance shows that the given knowledge of the monitor help the agent focus its exploration on state-actions that their environment rewards is observable even if the probability is low.

## C. Ablation Studies

In this section, we show that our innovations are crucial to extend MBIE-EB to Mon-MDPs. We show that without all our proposed innovations, there exists at least one setting that the resulting algorithm without the full innovations fails.

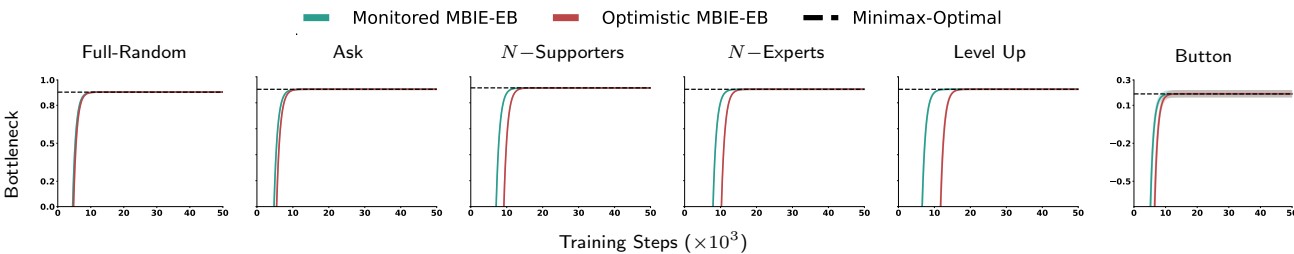

(a) **Comparison between Monitored MBIE-EB and MBIE-EB in solvable Bottleneck**. When all the rewards are observable, MBIE-EB's optimism is effective to learn all the unknown quantities. MBIE-EB matches the performance of Monitored MBIE-EB.

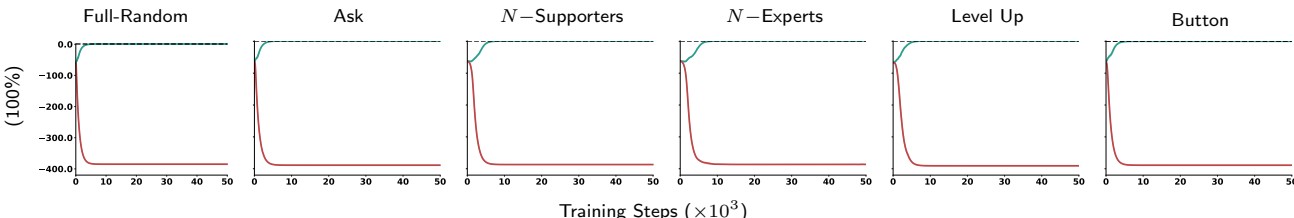

(b) **Comparison between Monitored MBIE-EB and MBIE-EB in unsolvable Bottleneck**. While Monitored MBIE-EB is pessimistic with respect to unobservable rewards, MBIE-EB remains mistakenly optimistic. MBIE-EB's optimism leads to endless visits to $\perp$ cells.

Figure 12: **Verifying the importance of the second innovation: pessimism instead of optimism.**

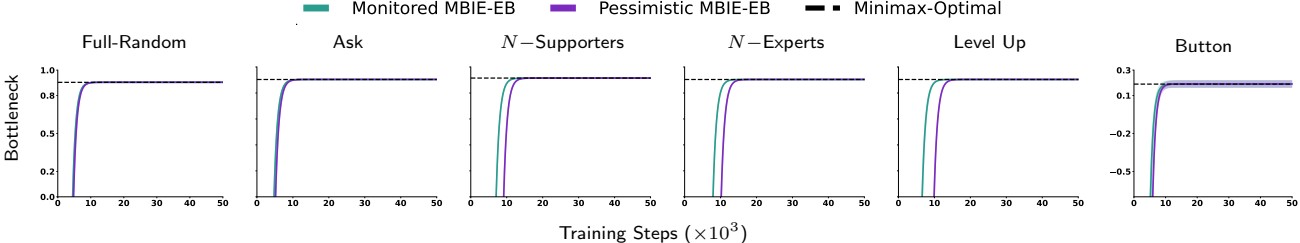

(a) **Comparison between Monitored MBIE-EB and pessimistic MBIE-EB in solvable Bottleneck**. When all the rewards are deterministically observable, pessimistic MBIE-EB is effective because a single visit to every state-action is sufficient to conclude that the reward is observable. Therefore, pessimistic MBIE-EB matches the performance of Monitored MBIE-EB.

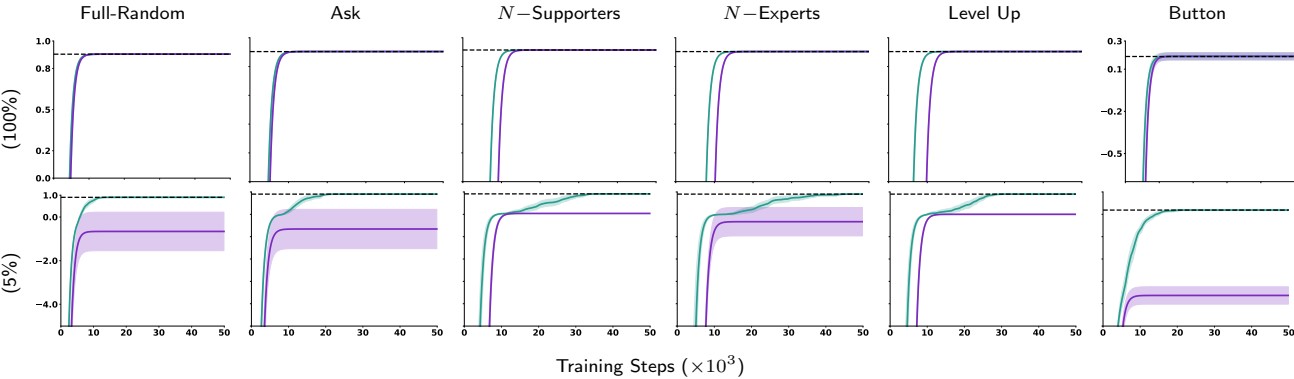

(b) **Comparison between Monitored MBIE-EB and pessimistic MBIE-EB in unsolvable Bottleneck**. With deterministic observability, pessimistic MBIE-EB is effective in finding the minimax-optimality. But with stochastic observability, pessimistic MBIE-EB due to premature pessimism with respect to state-actions that their reward can be observed upon enough exploration fails. On the other hand, due to exploring to observe the reward, Monitored MBIE-EB is robust against the stochasticity in the observability of the rewards.

Figure 13: **Verifying the importance of the third innovation: explore to observe rewards.**

### C.1. Importance of the Second Innovation: Pessimism Instead of Optimism

MBIE-EB uses the optimistic initial action-values for state-actions that their counts are zero. That is, in Mon-MDPs for any joint state-action pairs $(s, a) \equiv (s^{\text{e}}, s^{\text{m}}, a^{\text{e}}, a^{\text{m}})$ that any of $N(s^{\text{e}}, a^{\text{e}}), N(s^{\text{m}}, a^{\text{m}})$, or $N(s, a)$ is zero, $Q(s, a)$ would

be shortcut to an optimistic value. This approach contrasts with the Equation (6)'s pessimism, when $N(s^\text{e}, a^\text{e})$ is zero. Therefore, first we show that on Bottleneck environment, when the reward of all $\perp$ cells is observable, MBIE-EB is effective because upon enough visitation resulting from the optimism, the underlying environment reward will finally be observed. Figure 12a shows the MBIE-EB's efficacy in solvable Mon-MDPs. However, we then show the lack of necessary pessimism when the reward of all $\perp$ cells is unobservable make MBIE-EB ineffective. The MBIE-EB's inefficacy in unsolvable Mon-MDPs is due to the fact that the optimism never decreases, hence the agent would continue visiting state-actions with unobservable rewards for ever. Figure 12b shows the MBIE-EB's failure in the unsolvable Bottleneck.

### C.2. Importance of the Third Innovation: Explore to Observe Rewards.

Appendix C.1 showed the excessive optimism of MBIE-EB hinders its performance in unsolvable Mon-MDPs. Now, we demonstrate without exploring to observe rewards, using pessimism alone is still prone to failure. We extend MBIE-EB to Mon-MDPs and for all state-actions $(s, a) \equiv (s^\text{e}, s^\text{m}, a^\text{e}, a^\text{m})$, when $N(s^\text{e}, a^\text{e})$ is zero, we use $r^\text{e}_\text{min}$. This approach is effective in Mon-MDPs with deterministic observability ($\rho = 1$). Figure 13a shows the efficacy of pessimistic MBIE-EB in solvable Bottleneck with deterministic observability. Pessimistic MBIE-EB is also effective in *unsolvable* Mon-MDPs with deterministic observability, where only one visit to each state-action concludes whether the environment reward is observable or not. Figure 13b, (100%) row verifies this claim for pessimistic MBIE-EB on unsolvable Bottleneck with 100% observability. However, if the observability is stochastic then premature pessimism hinders the pessimistic MBIE-EB's performance because the algorithm has become pessimistic with respect to state-actions that otherwise could have observed their rewards eventually. This shortcoming of the pessimistic MBIE-EB compared to Monitored MBIE-EB that explore to observe the rewards and then uses pessimism *with high confidence* is evident in Figure 13b, (5%) row.

## D. Pseudocode

---
**Algorithm 1** Monitored MBIE-EB
---

**Require:** $\widetilde{Q}_\text{opt-init}, \widetilde{Q}_\text{obs-init}$

1:  $\widetilde{Q} \leftarrow 0$
2:  $\kappa \leftarrow 0$
3:  **for** episodes $k \coloneqq 1, 2, 3, \ldots$ **do**
4:     **if** $\kappa \leq \kappa^*(k)$ **then**
5:        *// Observe Episode*
6:        Compute $\widetilde{Q}_{obs}$ by performing 50 steps of value iteration on Equation (8), or use $\widetilde{Q}_\text{obs-init}$ if counts are zero.
7:        $\widetilde{Q} \leftarrow \widetilde{Q}_{obs}$
8:        $\kappa \leftarrow \kappa + 1$
9:     **else**
10:       *// Optimize Episode*
11:       Compute $\widetilde{Q}_{opt}$ by performing 50 steps of value iteration on Equation (6), or use $\widetilde{Q}_\text{opt-init}$ if counts are zero.
12:       $\widetilde{Q} \leftarrow \widetilde{Q}_{opt}$
13:     **end if**
14:     **for** steps $h \coloneqq 1, 2, \ldots, H$ **do**
15:       Follow the greedy policy with respect to $\widetilde{Q}$.
16:     **end for**
17: **end for**

---

## E. Low-Level Implementation Specifics

The minimum non-zero probability of observing the reward $\rho$ is by default one unless otherwise states. In experiments that include $N$-Supporters or $N$-Experts, $N$ is equal to four, and the number of levels for Level-Up is three. Hyperparameters of Directed-E$^2$ consist of: $Q_\text{int}$ the initial action-values, $\Psi_0$ the initial visitation-values, $r_0$ the initial values of the environment reward model, $\bar{\beta}$ goal-conditioned threshold specifying when a joint state-action pair should be visited using visitation-values, $\alpha$ the learning rate to update each $Q$ or $S$ incrementally, and discount factor $\gamma$ that is held fixed to 0.99. These values are directly reported from Parisi et al. (2024a). Monitored MBIE-EB's hyperparameters are set per environment and do not change across monitors. The same applies to Directed-E$^2$, but Parisi et al. (2024a) recommended to anneal the learning rate

---

**Algorithm 2** Directed-$E^2$

---

**Require:** $Q_{\text{int}}, \Psi_{\text{int}}$
1: $Q \leftarrow Q_{\text{int}}, \Psi \leftarrow \Psi_{\text{int}}$
2: $t \leftarrow 0$ // *Total timesteps*
3: **for** episodes $k := 1, 2, \ldots$ **do**
4:     **for** steps $h := 1, 2, \ldots$ **do**
5:         $(s^{\text{g}}, a^{\text{g}}) \leftarrow \arg\min_{s,a} N(s, a)$
6:         $\beta_t \leftarrow \log t / N(s^{\text{g}}, a^{\text{g}})$
7:         **if** $\beta_t > \bar{\beta}$ **then** $A_h \leftarrow \arg\max_a \Psi(a \mid S_h, s^{\text{g}}, a^{\text{g}})$ // *Explore*
8:         **else** $A_h \leftarrow \arg\max_a Q(S_h, a)$ // *Exploit*
9:         $t := t + 1$
10:       Perform action $A_h$
11:     **end for**
12: **end for**

---

for $N$-Supporters and $N$-Experts. KL-UCB has no closed form solution. We computed it using the Newton's method. The stopping condition for the Newton's method was chosen 50 iterations or the accuracy of at least $10^{-5}$ between successive solutions, which one happened first. We ran all the experiments on a SLURM-based cluster, using 32 Intel E5-2683 v4 Broadwell @ 2.1GHz CPUs. Thirty parallel runs took about an hour on a 32 core CPU.

### E.1. Hyperparameters

Let $\mathcal{U}$ denote the uniform distribution and $x \mapsto y$ the linear annealing from $x$ to $y$. Table 1 lists the hyperparameters used.

Table 1: The set of hyperparameters.

(a) **Hyperparameters of Monitored MBIE-EB across experiments.**

| | | Unknown monitor | | | | |
|---|---|---|---|---|---|---|
| Experiment | Environment | $\widetilde{Q}_{\text{opt-init}}$ | $\widetilde{Q}_{\text{obs-init}}$ | $\kappa^*(k)$ | $\beta^{\text{KL-UCB}}$ | $\beta^{\text{obs}}, \beta, \beta^{\text{m}}, \beta^{\text{e}}$ |
| Figure 6 | **Empty** | 1 | 100 | $\log_{1.005} k$ | $5 \times 10^{-2}$ | $5 \times 10^{-4}$ |
| | **Hazard** | 1 | 100 | $\log_{1.005} k$ | $5 \times 10^{-2}$ | $5 \times 10^{-4}$ |
| | **One-Way** | 1 | 100 | $\log_{1.005} k$ | $5 \times 10^{-2}$ | $5 \times 10^{-4}$ |
| | **River-Swim** | 30 | 100 | $\log_{1.005} k$ | $5 \times 10^{-2}$ | $5 \times 10^{-4}$ |
| Appendix B.3 | **Bottleneck** | 1 | 100 | $\log_{1.005} k$ | $5 \times 10^{-2}$ | $5 \times 10^{-4}$ |
| Appendix B.4 | **Bottleneck** | 1 | 100 | $\log_{1.005} k$ | $5 \times 10^{-2}$ | $5 \times 10^{-4}$ |

| | | Known monitor | | | | |
|---|---|---|---|---|---|---|
| Experiment | Environment | $\widetilde{Q}_{\text{opt-init}}$ | $\widetilde{Q}_{\text{obs-init}}$ | $\kappa^*(k)$ | $\beta^{\text{e}}$ | $\beta$ |
| Appendix B.5 | **Bottleneck** | 1 | 100 | $\log_{1.005} k$ | $5 \times 10^{-4}$ | $5 \times 10^{-4}$ |

(b) **Hyperparameters of MBIE-EB across experiments.**

| Experiment | Environment | $\widetilde{Q}_{\text{opt-init}}$ | $\beta, \beta^{\text{m}}, \beta^{\text{e}}$ |
|---|---|---|---|
| Appendix C | **Bottleneck** | 50 | $5 \times 10^{-4}$ |

(c) **Hyperparameters of Directed-$E^2$ across experiments.**

| | | (Annealing for $N$-Supporters and $N$-Experts) | | | | |
|---|---|---|---|---|---|---|
| Experiment | Environment | $Q_{\text{int}}$ | $\Psi_{\text{int}}$ | $r_0$ | $\bar{\beta}$ | $\alpha$ |
| Figure 6 | **Empty** | -10 | 1 | $\mathcal{U}[-0.1, 0.1]$ | $10^{-2}$ | $1(1 \mapsto 0.1)$ |
| | **One-Way** | -10 | 1 | $\mathcal{U}[-0.1, 0.1]$ | $10^{-2}$ | $1(1 \mapsto 0.1)$ |
| | **Hazard** | -10 | 1 | $\mathcal{U}[-0.1, 0.1]$ | $10^{-2}$ | $0.5(0.5 \mapsto 0.1)$ |
| | **River-Swim** | -10 | 1 | $\mathcal{U}[-0.1, 0.1]$ | $10^{-2}$ | $0.5 \mapsto 0.05$ |
| Appendix B.4 | **Bottleneck** | -10 | 1 | $-10$ | $10^{-2}$ | $1(1 \mapsto 0.1)$ |
| Appendix B.4 | **Bottleneck** | -10 | 1 | $-10$ | $10^{-2}$ | $1(1 \mapsto 0.1)$ |

# F. Auxiliary Propositions

**Lemma F.1.** *For any $a, x > 0$, $x \geq 2a \ln a$ implies $x \geq a \ln x$.*

*Proof.* First we prove that for $\forall x > 0$: $x > 2 \ln x$. Consider the function $y = x - 2 \ln x$. It is enough to prove that $y > 0$ is always true on its domain. we have that

$$\frac{dy}{dx} = 1 - \frac{2}{x}, \qquad \frac{d^2 y}{d^2 x} = \frac{2}{x^2} > 0.$$

Therefore, $y = x - 2 \ln x$ is convex. Also by

$$\frac{dy}{dx} = 1 - \frac{2}{x} = 0 \Rightarrow x = 2,$$

the minimum of $y = x - 2 \ln x = 2 - 2 \ln 2 > 0$. Thus, $x > 2 \ln x, \forall x > 0$. Back to the inequalities in the lemma, let $z = \frac{x}{a}$, then we need to prove that $z \geq 2 \ln a$ implies $z \geq \ln (z \cdot a)$.

- If $a \geq z$, then
$$\ln (z \cdot a) = \ln z + \ln a \leq 2 \ln a \leq z.$$
  Which is by the assumption of $z \geq 2 \ln a$.

- If $a < z$, then
$$\ln a < \ln z \Rightarrow \ln (z \cdot a) \leq 2 \ln z < z.$$
  which is by our proof in the beginning, where we showed $z \geq 2 \ln z, \forall z > 0$. $\qquad\square$

**Lemma F.2.** *Let $\Omega$ be an outcome space, and each of $(X_i)_{i=1}^n$ and $(Y_i)_{i=1}^n$ be $n$ random variables on $\Omega$. It holds that:*

$$\left\{ \sum_{i=1}^n X_i \geq \sum_{i=1}^n Y_i \right\} \subseteq \left\{ \bigcup_{i=1}^n (X_i \geq Y_i) \right\}.$$

*Proof.* Proof by contradiction.

Suppose $\{ \sum_{i=1}^n X_i \geq \sum_{i=1}^n Y_i \} \supset \{ \bigcup_{i=1}^n (X_i \geq Y_i) \}$, then there exists an $\omega \in \Omega$ that $\sum_{i=1}^n X_i(\omega) \geq \sum_{i=1}^n Y_i(\omega)$ but $X_1(\omega) < Y_1(\omega), X_2(\omega) < Y_2(\omega), \cdots X_n(\omega) < Y_n(\omega)$ resulting in $\sum_{i=1}^n X_i(\omega) < \sum_{i=1}^n Y_i(\omega)$ which is a contradiction. $\qquad\square$

**Corollary F.3.** *Let $(X_i)_{i=1}^n$ and $(Y_i)_{i=1}^n$ be $n$ random variables on probability space $(\Omega, \mathcal{F}, \mathbb{P})$. It holds that:*

$$\mathbb{P}\left( \sum_{i=1}^n X_i \geq \sum_{i=1}^n Y_i \right) \leq \sum_{i=1}^n \mathbb{P}(X_i \geq Y_i).$$

*Proof.* Using Lemma F.2 and due to monotonicity of measures we have

$$\mathbb{P}\left( \sum_{i=1}^n X_i \geq \sum_{i=1}^n Y_i \right) \leq \mathbb{P}\left( \bigcup_{i=1}^n (X_i \geq Y_i) \right).$$

By applying the union bound the inequality is obtained. $\qquad\square$

**Lemma F.4.** *Let $M_1 = \langle \mathcal{S}, \mathcal{A}, r_1, p_1, f^m, \gamma \rangle$ and $M_2 = \langle \mathcal{S}, \mathcal{A}, r_2, p_2, f^m, \gamma \rangle$ be two Mon-MDPs. Assume that*

$$-r_{max}^e \leq r_1^e, r_2^e \leq r_{max}^e \qquad and \qquad -r_{max}^m \leq r_1^m, r_2^m \leq r_{max}^m.$$

*Additionally assume that for all joint state-action pairs $(s, a) \equiv (s^e, s^m, a^e, a^m)$, it holds that*

$$|r_1^e(s^e, a^e) - r_2^e(s^e, a^e)| \leq \varphi^e, \qquad |r_1^m(s^m, a^m) - r_2^m(s^m, a^m)| \leq \varphi^m, \qquad \|p_1(\cdot \mid s, a) - p_2(\cdot \mid s, a)\|_1 \leq \varphi.$$

*Then for any stationary deterministic policies $\pi$, it holds that*

$$|Q_1^\pi(s, a) - Q_2^\pi(s, a)| \leq \frac{\varphi^e + \varphi^m + 2\varphi\gamma(r_{max}^e + r_{max}^m)}{(1 - \gamma)^2}.$$

*Proof.* Let

$$\Delta := \max_{(s,a)} |Q_1^\pi(s,a) - Q_2^\pi(s,a)|, \quad \Delta^{\mathrm{e}} := r_1^{\mathrm{e}}(s^{\mathrm{e}}, a^{\mathrm{e}}) - r_2^{\mathrm{e}}(s^{\mathrm{e}}, a^{\mathrm{e}}), \quad \Delta^{\mathrm{m}} := r_1^{\mathrm{m}}(s^{\mathrm{m}}, a^{\mathrm{m}}) - r_2^{\mathrm{m}}(s^{\mathrm{m}}, a^{\mathrm{m}}),$$

$$\Delta^{\mathrm{p}} := \gamma \sum_{s'} p_1(s'|s,a) V_1^\pi(s') - \gamma \sum_{s'} p_2(s'|s,a) V_2^\pi(s').$$

Then

$$|Q_1^\pi(s,a) - Q_2^\pi(s,a)| = |\Delta_1 + \Delta^{\mathrm{m}} + \Delta^{\mathrm{p}}| \leq |\Delta_1| + |\Delta^{\mathrm{m}}| + |\Delta^{\mathrm{p}}| \qquad \text{(Triangle inequality)}$$

$$= \varphi^{\mathrm{e}} + \varphi^{\mathrm{m}} + \gamma \left| \sum_{s'} (p_1(s'|s,a) V_1^\pi(s') - p_2(s'|s,a) V_2^\pi(s')) \right|$$

$$= \varphi^{\mathrm{e}} + \varphi^{\mathrm{m}} + \gamma \left| \sum_{s'} (p_1(s'|s,a) V_1^\pi(s') - p_1(s'|s,a) V_2^\pi(s') + p_1(s'|s,a) V_2^\pi(s') - p_2(s'|s,a) V_2^\pi(s')) \right|$$

$$= \varphi^{\mathrm{e}} + \varphi^{\mathrm{m}} + \gamma \left| \sum_{s'} (p_1(s'|s,a)(V_1^\pi(s') - V_2^\pi(s')) + (p_1(s'|s,a) - p_2(s'|s,a)) V_2^\pi(s')) \right|$$

$$\leq \varphi^{\mathrm{e}} + \varphi^{\mathrm{m}} + \gamma \left| \sum_{s'} p_1(s'|s,a)(V_1^\pi(s') - V_2^\pi(s')) \right| + \gamma \left| \sum_{s'} (p_1(s'|s,a) - p_2(s'|s,a)) V_2^\pi(s') \right|$$

$$\leq \varphi^{\mathrm{e}} + \varphi^{\mathrm{m}} + \gamma \left| \sum_{s'} p_1(s'|s,a)(V_1^\pi(s') - V_2^\pi(s')) \right| + \frac{2\gamma\varphi(r_{\max}^{\mathrm{e}} + r_{\max}^{\mathrm{m}})}{1 - \gamma}.$$

By taking the $\max_{(s,a)}$ from the both sides we have

$$\Delta \leq \varphi^{\mathrm{e}} + \varphi^{\mathrm{m}} + \gamma\Delta + \frac{2\gamma\varphi(r_{\max}^{\mathrm{e}} + r_{\max}^{\mathrm{m}})}{1 - \gamma}$$

$$(1 - \gamma)\Delta \leq \varphi^{\mathrm{e}} + \varphi^{\mathrm{m}} + \frac{2\gamma\varphi(r_{\max}^{\mathrm{e}} + r_{\max}^{\mathrm{m}})}{1 - \gamma}$$

$$\Delta \leq \frac{\varphi^{\mathrm{e}} + \varphi^{\mathrm{m}}}{1 - \gamma} + \frac{2\gamma\varphi(r_{\max}^{\mathrm{e}} + r_{\max}^{\mathrm{m}})}{(1 - \gamma)^2} \leq \frac{\varphi^{\mathrm{e}} + \varphi^{\mathrm{m}} + 2\gamma\varphi(r_{\max}^{\mathrm{e}} + r_{\max}^{\mathrm{m}})}{(1 - \gamma)^2}.$$

Since $|Q_1^\pi(s,a) - Q_2^\pi(s,a)| \leq \Delta$, the proof is completed. □

**Lemma F.5.** *Let $M_1 = \langle \mathcal{S}, \mathcal{A}, r_1, p_1, f^m, \gamma \rangle$ and $M_2 = \langle \mathcal{S}, \mathcal{A}, r_2, p_2, f^m, \gamma \rangle$ be two Mon-MDPs. Assume that:*

$$-r_{max}^e \leq r_1^e, r_2^e \leq r_{max}^e \qquad and \qquad -r_{max}^m \leq r_1^m, r_2^m \leq r_{max}^m.$$

*Suppose further that for all joint state-action pairs $(s,a) \equiv (s^e, s^m, a^e, a^m)$, it holds*

$$|r_1^e(s^e, a^e) - r_2^e(s^e, a^e)| \leq \varphi^e, \qquad |r_1^m(s^m, a^m) - r_2^m(s^m, a^m)| \leq \varphi^m, \qquad \|p_1(\cdot \mid s,a) - p_2(\cdot \mid s,a)\|_1 \leq \varphi.$$

*There exists a constant $C$ that for any $0 \leq \varepsilon \leq \frac{(r_{max}^e + r_{max}^m)}{1 - \gamma}$, and any stationary policy $\pi$, if $\varphi^e = \varphi^m = \varphi = C\frac{\varepsilon(1-\gamma)^2}{r_{max}^e + r_{max}^m}$, then*

$$|Q_1^\pi(s,a) - Q_2^\pi(s,a)| \leq \varepsilon.$$

*Proof.* Using Lemma F.4, we show that $\frac{\varphi^{\mathrm{e}} + \varphi^{\mathrm{m}} + 2\varphi\gamma(r_{\max}^{\mathrm{e}} + r_{\max}^{\mathrm{m}})}{(1-\gamma)^2} = 2\varphi \frac{1 + \gamma(r_{\max}^{\mathrm{e}} + r_{\max}^{\mathrm{m}})}{(1-\gamma)^2} \leq \varepsilon$, which yields $\varphi \leq \frac{\varepsilon(1-\gamma)^2}{2(1 + \gamma(r_{\max}^{\mathrm{e}} + r_{\max}^{\mathrm{m}}))}$. By assumption that $r_{\max}^{\mathrm{e}} = r_{\max}^{\mathrm{m}} = 1$ and choosing $C = \frac{1}{6}$, we have $\varphi = \frac{\varepsilon(1-\gamma)^2}{6} \leq \frac{\varepsilon(1-\gamma)^2}{2(1 + \gamma(r_{\max}^{\mathrm{e}} + r_{\max}^{\mathrm{m}}))}$. □

**Lemma F.6** (Chernoff-Hoeffding's inequality). *For $(X_i)_{i=1}^n$ independent identically distributed samples on probability space $(\Omega, \mathcal{F}, \mathbb{P})$, where $X_i \in [a_i, b_i]$ for all $i$ and $\epsilon > 0$, we have*

$$\mathbb{P}\left(\mathbb{E}[X_1] - \frac{1}{n}\sum_{i=1}^n X_i \geq \epsilon\right) \leq \exp\left(-\frac{2n^2\epsilon^2}{\sum_{i=1}^n(b_i - a_i)^2}\right).$$

# G. Proof of Theorem 3.1

The complete proof essentially follows all the steps of Strehl & Littman (2008, Theorem 2) with additional modifications. We only provide proofs for components that are non-trivially distinct between MDPs and Mon-MDPs when the analysis of Strehl & Littman (2008, Theorem 2) is applied. Essentially, Theorem 3.1 comprises five high probability statements:

1. Specifying the number of samples required to estimate the observability of the environment reward in observe episodes

2. Optimism of $\widetilde{Q}_{\text{obs}}$

3. Specifying the number of samples required to find a near minimax-optimal policy in optimize episodes

4. Optimism of $\widetilde{Q}_{\text{opt}}$

5. Determining the sample complexity

For the first two steps of the proof let

$$b(s,a) = \mathbb{E}\left[\mathbb{1}\left\{\widehat{R}_{t+1}^{\text{e}} \neq \perp\right\} \cdot \mathbb{1}\left\{N(S_t^{\text{e}}, A_t^{\text{e}}) = 0\right\}\Big| S_t = s, A_t = a\right],$$

denote the expected observability of the environment reward for a joint state-action $(s, a)$ that its environment reward has not been observed until timestep $t$. Note that the KL-UCB term in Equation (8) is an upper confidence bound on $b$. Also, define $\bar{B}$ to be the maximum likelihood estimation of $b$.

## G.1. Specifying The Number of Samples Required to Estimate The Observability of Reward in Observe Episodes

According to Strehl & Littman (2008, Lemma 2), to find a policy in observe episodes that its action-values, computed using $\bar{B}$ and $\bar{P}$, are $\varepsilon_1$-accurate, $\bar{B}$ and $\bar{P}$ should be $\tau$-close to their true mean for all state-actions $(s, a)$, where $\tau = \frac{\varepsilon_1(1-\gamma)^2}{2}$. Hence, to specify the least number of visits $m_1$ to $(s, a)$ to fulfill the $\tau$-closeness, we must have

$$\left\|\bar{P}(\cdot \mid s, a) - p(\cdot \mid s, a)\right\|_1 \leq \tau, \tag{10}$$

$$\left|\bar{B}(s, a) - b(s, a)\right| \leq \tau. \tag{11}$$

Also, according to Strehl & Littman (2008) if $(s, a)$ has been visited $m_1$ times, with probabilities at least $1 - \delta^p$ and $1 - \delta^b$,

$$\left\|\bar{P}(\cdot \mid s, a) - p(\cdot \mid s, a)\right\|_1 \leq \sqrt{\frac{2\left[\ln\left(2^{|\mathcal{S}|} - 2\right) - \ln \delta^p\right]}{m_1}}, \qquad \left|\bar{B}(s, a) - b(s, a)\right| \leq \sqrt{\frac{\ln \frac{2}{\delta^b}}{2m_1}}.$$

Hence, for Equations (10) and (11) to hold simultaneously with probability at $1 - \delta_1$ until $(s, a)$ is visited $m_1$ times, by setting $\delta^p = \delta^b = \frac{\delta_1}{2|\mathcal{S}||\mathcal{A}|m_1}$ to split the failure probability equally for all state-actions until each of them has been visited $m_1$ times, it is enough to ensure $\tau$ is bigger than the length of the confidence intervals:

$$m_1 \geq \max\left\{\frac{8\left[\ln\left(2^{|\mathcal{S}|} - 2\right) - \ln \delta^p\right]}{\tau^2}, \frac{2\ln \frac{2}{\delta^b}}{\tau^2}\right\} \geq \max\left\{\frac{8\left[\ln\left(2^{|\mathcal{S}|} - 2\right) - \ln \frac{\delta_1}{2|\mathcal{S}||\mathcal{A}|m_1}\right]}{\tau^2}, \frac{2\ln \frac{4|\mathcal{S}||\mathcal{A}|m_1}{\delta_1}}{\tau^2}\right\}$$

$$= \frac{8\left[\ln\left(2^{|\mathcal{S}|} - 2\right) + \ln \frac{2|\mathcal{S}||\mathcal{A}|m_1}{\delta_1}\right]}{\tau^2}. \tag{12}$$

Lemma F.1 helps us to bring $m_1$ out of the right-hand side of Equation (12). Hence,

$$m_1 = \mathcal{O}\left(\frac{|\mathcal{S}|}{\tau^2} + \frac{1}{\tau^2} \ln \frac{|\mathcal{S}||\mathcal{A}|}{\tau \delta_1}\right) = \mathcal{O}\left(\frac{|\mathcal{S}|}{\varepsilon_1^2(1-\gamma)^4} + \frac{1}{\varepsilon_1^2(1-\gamma)^4} \ln \frac{|\mathcal{S}||\mathcal{A}|}{\varepsilon_1(1-\gamma)^2 \delta_1}\right). \tag{13}$$

Equation (13) shows the fact that regardless of how infinitesimal the probability of observing the reward is, in the worst-case the difficulty of learning expected observability lies in learning the transition dynamics (Kakade, 2003; Szita & Szepesvári, 2010), i.e., by the time the transition dynamics are approximately learned, the agent has approximately figured out what the probability of observing the environment reward for taking any joint action is.

### G.2. Optimism of $\widetilde{Q}_{\text{obs}}$

Optimism is needed to make sure the agent has enough incentives to visit state-actions with inaccurate statistics. Suppose $m_1$ is the least number of samples required to ensure $\bar{B}$ and $\bar{P}$ are accurate. Therefore, the agent should be optimistic about the state-actions that have not been visited $m_1$ times yet. Let

$$Q_{\text{obs}}^*(s,a) = b(s,a) + \gamma \sum_{s'} p(s' \mid s,a) \max_{a'} Q_{\text{obs}}^*(s',a').$$

Now consider the first $v$ visits to $(s,a)$ during observe episodes, where $v < m_1$ and consider $\widetilde{Q}_{\text{obs}}$ as

$$\widetilde{Q}_{\text{obs}}(s,a) = \bar{B}(s,a) + \gamma \sum_{s'} \bar{P}(s' \mid s,a) \max_{a'} \widetilde{Q}_{\text{obs}}(s',a') + \frac{\beta^{\text{obs}}}{\sqrt{v}}. \tag{14}$$

But, $\bar{B}(s,a)$ is zero for all state-action pairs. Because if $\bar{B}(s,a)$ is not zero, then $\mathbb{1}\left\{N(S_t^{\text{e}}, A_t^{\text{e}}) = 0 | S_t = s, A_t = a\right\}$ must return one, which means environment reward of $(s,a)$ has not been observed until timestep $t$. Consequently, $\left\{\widehat{R}_{t+1}^{\text{e}} \neq \perp \Big| S_t = s, A_t = a\right\}$ has been zero until timestep $t$. Therefore, Equation (14) turns into

$$\widetilde{Q}_{\text{obs}}(s,a) = \gamma \sum_{s'} \bar{P}(s' \mid s,a) \max_{a'} \widetilde{Q}_{\text{obs}}(s',a') + \frac{\beta^{\text{obs}}}{\sqrt{v}}. \tag{15}$$

According to Strehl & Littman (2008, Lemma 7), by choosing $\beta^{\text{obs}} = (1-\gamma)^{-1}\sqrt{0.5\ln\left(\frac{4|\mathcal{S}||\mathcal{A}|m_1}{\delta_1}\right)}$,

$$\widetilde{Q}_{\text{obs}}(s,a) = \gamma \sum_{s'} \bar{P}(s' \mid s,a) \max_{a'} \widetilde{Q}_{\text{obs}}(s',a') + \frac{\beta^{\text{obs}}}{\sqrt{v}} \geq Q_{\text{obs}}^*(s,a),$$

with probability at least $1 - \frac{\delta_1}{4|\mathcal{S}||\mathcal{A}|m_1}$. On the other hand by choosing $\beta^{\text{KL-UCB}} = \ln(4|\mathcal{S}||\mathcal{A}|m_1\delta_1)$,

$$\text{KL-UCB}(v) = \max\{\mu \in [0,1] : d(0,\mu) \leq \frac{\beta^{\text{KL-UCB}}}{v}\} \geq b(s,a),$$

with probability at least $1 - \frac{\delta_1}{4|\mathcal{S}||\mathcal{A}|m_1}$. Now define the random variables $X_1$ and $X_2$ as

$$X_1 := Q_{\text{obs}}^*(s,a) - \gamma \sum_{s'} \bar{P}(s' \mid s,a) \max_{a'} \widetilde{Q}_{\text{obs}}(s',a'), \qquad X_2 := \bar{B}(s,a) = 0.$$

We have:

$$\mathbb{P}\left(X_1 \geq \underbrace{\frac{\beta^{\text{obs}}}{\sqrt{v}}}_{Y_1}\right) \leq \frac{\delta_1}{4|\mathcal{S}||\mathcal{A}|m_1}, \qquad \mathbb{P}\left(X_2 \geq \underbrace{\max\left\{\mu \in [0,1] : d(0,\mu) \leq \frac{\beta^{\text{KL-UCB}}}{v}\right\}}_{Y_2}\right) \leq \frac{\delta_1}{4|\mathcal{S}||\mathcal{A}|m_1}$$

Using Corollary F.3 we have:

$$\mathbb{P}(X_1 + X_2 \geq Y_1 + Y_2) \leq \frac{\delta_1}{2|\mathcal{S}||\mathcal{A}|m_1}.$$

Therefore, with probability at least $1 - \frac{\delta_1}{2|\mathcal{S}||\mathcal{A}|m_1}$ we must have that $X_1 + X_2 \leq Y_1 + Y_2$. Thus,

$$Q_{\text{obs}}^*(s,a) - \gamma \sum_{s'} \bar{P}(s' \mid s,a) \max_{a'} \widetilde{Q}_{\text{obs}}(s',a') \leq Y_1 + Y_2$$

$$Q_{\text{obs}}^*(s,a) \leq \gamma \sum_{s'} \bar{P}(s' \mid s,a) \max_{a'} \widetilde{Q}_{\text{obs}}(s',a') + Y_1 + Y_2$$

$$Q_{\text{obs}}^*(s,a) \leq \widetilde{Q}_{\text{obs}}(s,a) + \max\left\{\mu \in [0,1] : d(0,\mu) \leq \frac{\beta^{\text{KL-UCB}}}{v}\right\}.$$

By abusing the notation for $\widetilde{Q}_{\text{obs}}$ to incorporate the KL-UCB term, we have $Q_{\text{obs}}^*(s,a) \leq \widetilde{Q}_{\text{obs}}(s,a)$, which by the union bound, with probability at least $1 - \frac{\delta_1}{2}$, for all state-actions holds until they are visited at least $m_1$ times.

### G.3. Specifying The Number of Samples Required to Find A Near-Minimax-Optimal Policy in Optimize Episodes

During optimize episodes the agent can face two types of joint state-actions: 1) state-actions that lead to observing the environment reward, e.g., moving and asking for reward 2) state-actions that do not lead to observing the environment reward e.g., moving and not asking for reward. Let denote these sets as the **observable** and the **unobservable** respectively.

#### G.3.1. Number of Samples for The Observable Set

In contrast to Appendix G.1, which is an off-the-shelf application of the analysis done by Strehl & Littman (2008, Lemma 5), since in optimize episodes we have three unknown quantities $r^\mathrm{e}$, $r^\mathrm{m}$, and $p$ –that are mappings from different input spaces– we need Lemmas F.4 and F.5 that are straight adaptations of Strehl & Littman (2008, Lemma 1 and 2). By Lemma F.5, setting $\tau = \frac{\varepsilon_2(1-\gamma)^2}{6}$ ensures an $\varepsilon_2$-minimax-optimal policy for state-actions in the observable set. We have,

$$\left|\bar{R}^\mathrm{e}(s^\mathrm{e}, a^\mathrm{e}) - r^\mathrm{e}(s^\mathrm{e}, a^\mathrm{e})\right| \leq \tau, \qquad \left|\bar{R}^\mathrm{m}(s^\mathrm{m}, a^\mathrm{m}) - r^\mathrm{m}(s^\mathrm{m}, a^\mathrm{m})\right| \leq \tau, \qquad \left\|\bar{P}(\cdot \mid s, a) - p(\cdot \mid s, a)\right\|_1 \leq \tau.$$

On the other hand, if $(s, a) \equiv (s^\mathrm{e}, s^\mathrm{m}, a^\mathrm{e}, a^\mathrm{m})$ has been visited $v$ times, its monitor reward has been observed $v^\mathrm{m}$ times, and its environment reward has been observed $v^\mathrm{e}$ times, with probabilities at least $1 - \delta^\mathrm{e}$, $1 - \delta^\mathrm{m}$, and $1 - \delta^p$, it holds,

$$\left\|\bar{P}(\cdot \mid s, a) - p(\cdot \mid s, a)\right\|_1 \leq \sqrt{\frac{2\left[\ln\left(2^{|\mathcal{S}|} - 2\right) - \ln \delta^p\right]}{v}}, \tag{16}$$

$$\left|\bar{R}^\mathrm{m}(s^\mathrm{m}, a^\mathrm{m}) - r^\mathrm{m}(s^\mathrm{m}, a^\mathrm{m})\right| \leq \sqrt{\frac{2\ln(2/\delta^\mathrm{m})}{v^\mathrm{m}}}, \tag{17}$$

$$\left|\bar{R}^\mathrm{e}(s^\mathrm{e}, a^\mathrm{e}) - r^\mathrm{e}(s^\mathrm{e}, a^\mathrm{e})\right| \leq \sqrt{\frac{\ln(2/\delta^\mathrm{e})}{v^\mathrm{e}}}. \tag{18}$$

Therefore, to find $m_2$, the least required number of visits to $(s, a)$ in optimize episodes, we make connections among $m_2, v, v^\mathrm{m}$, and $v^\mathrm{e}$. If a joint state-action is visited $m_2$ times, then we have,

$$m_2 = v, \qquad m_2 \leq v^\mathrm{m} \leq \sum_{s^\mathrm{e} \in \mathcal{S}^\mathrm{e}} \sum_{a^\mathrm{e} \in \mathcal{A}^\mathrm{e}} m_2 = |\mathcal{S}^\mathrm{e}||\mathcal{A}^\mathrm{e}|m_2, \qquad m_2 \cdot \rho \leq v^\mathrm{e},$$

where $m_2 \cdot \rho \leq v^\mathrm{e}$ holds since the environment reward is observed with probability $\rho$ with each visit to $(s, a)$. To ensure Equations (16) to (18) hold with probability at least $1 - \delta_2$ until every $(s, a)$ is visited $m_2$ times, we set $\delta^p = \delta^\mathrm{m} = \frac{\delta_2}{3|\mathcal{S}||\mathcal{A}|m_2}$ and $\delta^\mathrm{e} = \frac{\delta_2}{3|\mathcal{S}||\mathcal{A}|\rho m_2}$, splitting failure probability evenly, and choose $\tau$ larger than the confidence intervals:

$$m_2 \geq \max \left\{ \frac{8\left[\ln\left(2^{|\mathcal{S}|} - 2\right) - \ln \delta^p\right]}{\tau^2}, \frac{8\ln\left(2/\delta^\mathrm{m}\right)}{\tau^2}, \frac{8\ln\left(2/\delta^\mathrm{e}\right)}{\tau^2} \right\}$$

$$\geq \max \left\{ \frac{8\left[\ln\left(2^{|\mathcal{S}|} - 2\right) - \ln \delta^p\right]}{\tau^2}, \frac{8\ln\left(2/\delta^\mathrm{e}\right)}{\rho\tau^2} \right\}$$

$$\geq \max \left\{ \frac{8\left[\ln\left(2^{|\mathcal{S}|} - 2\right) + \ln \frac{3|\mathcal{S}||\mathcal{A}|m_2}{\delta_2}\right]}{\tau^2}, \frac{8\ln \frac{6|\mathcal{S}||\mathcal{A}|\rho m_2}{\delta_2}}{\rho\tau^2} \right\}.$$

- If $\frac{1}{\rho} \geq \mathcal{O}(|\mathcal{S}|)$, then

$$m_2 \geq \frac{8\ln \frac{6|\mathcal{S}||\mathcal{A}|\rho m_2}{\delta_2}}{\rho\tau^2},$$

which by Lemma F.1 implies

$$m_2 = \mathcal{O}\left(\frac{1}{\rho\tau^2} \ln \frac{|\mathcal{S}||\mathcal{A}|}{\tau\delta_2}\right) = \mathcal{O}\left(\frac{1}{\rho\varepsilon_2^2(1-\gamma)^4} \ln \frac{|\mathcal{S}||\mathcal{A}|}{\varepsilon_2(1-\gamma)^2\delta_2}\right). \tag{19}$$

- If $\frac{1}{\rho} \leq \mathcal{O}(|\mathcal{S}|)$, then

$$m_2 \geq \frac{8 \left[ \ln \left( 2^{|\mathcal{S}|} - 2 \right) + \ln \frac{3|\mathcal{S}||\mathcal{A}| m_2}{\delta_2} \right]}{\tau^2}.$$

which by Lemma F.1 implies

$$m_2 = \mathcal{O} \left( \frac{|\mathcal{S}|}{\tau^2} + \frac{1}{\tau^2} \ln \frac{|\mathcal{S}| |\mathcal{A}|}{\tau \delta_2} \right) = \mathcal{O} \left( \frac{|\mathcal{S}|}{\varepsilon_2^2 (1 - \gamma)^4} + \frac{1}{\varepsilon_2^2 (1 - \gamma)^4} \ln \frac{|\mathcal{S}| |\mathcal{A}|}{\varepsilon_2 (1 - \gamma)^2 \delta_2} \right). \tag{20}$$

### 3.2 Number of Samples for The Unobservable Set

These state-actions cannot change the sample estimate of the environment reward and the only quantities updated by visiting them are the transition dynamics and the monitor reward. It is enough to have

$$m_2 \geq \max \left\{ \frac{8 \left[ \ln \left( 2^{|\mathcal{S}|} - 2 \right) - \ln \delta^p \right]}{\tau^2}, \frac{8 \ln \left( \frac{2}{\delta^{\mathrm{m}}} \right)}{\tau^2} \right\}.$$

Hence, similar to Appendix G.3.1, when $\frac{1}{\rho} \leq \mathcal{O}(|\mathcal{S}|)$, the dominant factor around learning the sample estimates is the transitions and the required sample size is

$$m_2 = \mathcal{O} \left( \frac{|\mathcal{S}|}{\varepsilon_2^2 (1 - \gamma)^4} + \frac{1}{\varepsilon_2^2 (1 - \gamma)^4} \ln \frac{|\mathcal{S}| |\mathcal{A}|}{\varepsilon_2 (1 - \gamma)^2 \delta_2} \right). \tag{21}$$

Therefore, overall the interplay between $\frac{1}{\rho}$ and $\mathcal{O}(|\mathcal{S}|)$ determines the value of $m_2$. In the worst-case $\frac{1}{\rho} \geq \mathcal{O}(|\mathcal{S}|)$ and

$$m_2 = \mathcal{O} \left( \frac{1}{\rho \varepsilon_2^2 (1 - \gamma)^4} \ln \frac{|\mathcal{S}| |\mathcal{A}|}{\varepsilon_2 (1 - \gamma)^2 \delta_2} \right). \tag{22}$$

## G.4. Optimism of $\widetilde{Q}_{\mathbf{opt}}$

Similar to observe episodes, optimism is needed to ensure the agent visits state-actions with inaccurate statistics. Let, $m_2$ be the least number of samples required to ensure $\bar{R}^{\mathrm{m}}$, $\bar{P}$, and if possible, $\bar{R}^{\mathrm{e}}$ are accurate. To be pessimistic in state-actions that $\bar{R}^{\mathrm{e}}$ cannot be computed due to their ever-lasting unobservability, we need to investigate the optimism in two cases where $\bar{R}^{\mathrm{e}}$ can be computed and when it cannot. Consider $v$ experiences of a joint state-action $(s, a) \equiv (s^{\mathrm{e}}, s^{\mathrm{m}}, a^{\mathrm{e}}, a^{\mathrm{m}})$ and the first $v^{\mathrm{e}}$ experiences of the environment state-action $(s^{\mathrm{e}}, a^{\mathrm{e}})$ in which the environment reward has been observed. Also, let $\bar{V}_{\downarrow}^*$ be

$$\bar{V}_{\downarrow}^*(s) := \max_a \bar{Q}_{\downarrow}^*(s, a) := \sum_{s'} \bar{P}(s' \mid s, a) V_{\downarrow}^*(s'), \quad \forall s \in \mathcal{S}.$$

### G.4.1. When $v^{\mathrm{E}} > 0$

Let $X_{1i}$, $X_{2i}$, and $X_{3i}$ be random variables defined at the $i$th visit, where $R_i^{\mathrm{e}}$ and $R_i^{\mathrm{m}}$ are the immediate environment and monitor reward at the $i$th visit and $S_i'$ is the next state visited after the the $i$th visit:

$$X_{1i} = R_i^{\mathrm{e}}, \quad X_{2i} = R_i^{\mathrm{m}}, \quad X_{3i} = \gamma \bar{V}_{\downarrow}^*(S_i').$$

If $(s, a)$ has been visited $v$ times and $R_i^{\mathrm{e}}$ has been observed $v^{\mathrm{e}} \leq v$ times, then: 1) the set $(X_{1i})_{i=1}^{v^{\mathrm{e}}}$ has been observed, 2) at least the set $(X_{2i})_{i=1}^{v}$ is available $\left( \text{at most } (X_{2i})_{i=1}^{|\mathcal{S}^{\mathrm{e}}||\mathcal{A}^{\mathrm{e}}| v} \right)$, 3) the set $(X_{3i})_{i=1}^{v}$ is available.

Define $(X_{1i})_{i=1}^{v^{\mathrm{e}}}$, $(X_{2i})_{i=1}^{v}$, and $(X_{3i})_{i=1}^{v}$ on joint probability space $(\Omega, \mathcal{F}, \mathbb{P})$. Using the Chernoff-Hoeffding's inequality:

- For $X_{1i}$ and $X_{2i}$ we have

$$\mathbb{P} \left( \mathbb{E}[X_{11}] - \frac{1}{v^{\mathrm{e}}} \sum_{i=1}^{v^{\mathrm{e}}} X_{1i} \geq Y_1 \right) \leq \exp \left( -\frac{v^{\mathrm{e}} Y_1^2}{2} \right), \quad \mathbb{P} \left( \mathbb{E}[X_{21}] - \frac{1}{v} \sum_{i=1}^{v} X_{2i} \geq Y_2 \right) \leq \exp \left( -\frac{v Y_2^2}{2} \right).$$

- For $X_{3i}$ we have that $\frac{-2\gamma}{1-\gamma} \le X_{3i} \le \frac{2\gamma}{1-\gamma}$, hence

$$\mathbb{P}\left(\mathbb{E}\left[X_{31}\right] - \frac{1}{v}\sum_{i=1}^{v} X_{3i} \ge Y_3\right) \le \exp\left(-\frac{vY_3^2(1-\gamma)^2}{8\gamma^2}\right).$$

Define $X_1, X_2,$ and $X_3$ on $(\Omega, \mathcal{F}, \mathbb{P})$ as

$$X_1 = \mathbb{E}\left[X_{11}\right] - \frac{1}{v^e}\sum_{i=1}^{v^e} X_{1i}, \quad X_2 = \mathbb{E}\left[X_{21}\right] - \frac{1}{v}\sum_{i=1}^{v} X_{2i}, \quad X_3 = \mathbb{E}\left[X_{31}\right] - \frac{1}{v}\sum_{i=1}^{v} X_{3i}.$$

By choosing $Y_1 = \frac{\beta^e}{\sqrt{v^e}}, Y_2 = \frac{\beta^m}{\sqrt{v}},$ and $Y_3 = \frac{\beta}{\sqrt{v}},$ where

$$\beta^e = \sqrt{2\ln\frac{6\,|\mathcal{S}|\,|\mathcal{A}|\,m_2}{\delta_2}}, \quad \beta^m = \sqrt{2\ln\frac{6\,|\mathcal{S}|\,|\mathcal{A}|\,m_2}{\delta_2}}, \quad \beta = \frac{2\gamma\sqrt{2\ln\left(6\,|\mathcal{S}|\,|\mathcal{A}|\,m_2/\delta_2\right)}}{1-\gamma},$$

and using Corollary F.3, we have

$$\mathbb{P}\left(X_1 + X_2 + X_3 \ge Y_1 + Y_2 + Y_3\right) \le \exp\left(-\frac{v^e Y_1^2}{2}\right) + \exp\left(-\frac{vY_2^2}{2}\right) + \exp\left(-\frac{vY_3^2(1-\gamma)^2}{8\gamma^2}\right).$$

Thus,

$$\mathbb{P}\left(X_1 + X_2 + X_3 \ge \frac{\beta^e}{\sqrt{v^e}} + \frac{\beta^m}{\sqrt{v}} + \frac{\beta}{\sqrt{v}}\right) \le \frac{\delta_2}{2\,|\mathcal{S}|\,|\mathcal{A}|\,m_2}. \tag{23}$$

Therefore, with probability at least $1 - \frac{\delta_2}{2|\mathcal{S}||\mathcal{A}|m_2}$ it must hold that $X_1 + X_2 + X_3 \le \left(\frac{\beta^e}{\sqrt{v^e}} + \frac{\beta^m}{\sqrt{v}} + \frac{\beta}{\sqrt{v}}\right)$, which is equal to

$$\frac{1}{v^e}\sum_{i=1}^{k} R_i^e + \frac{1}{v}\sum_{j=1}^{v}\left(R_j^m + \gamma\bar{V}_\downarrow^*(S_j')\right) + \left(\frac{\beta^e}{\sqrt{v^e}} + \frac{\beta^m}{\sqrt{v}} + \frac{\beta}{\sqrt{v}}\right) \ge \mathbb{E}\left[R_1^e + R_1^m + \gamma\bar{V}_\downarrow^*(S_1')\right] = Q_\downarrow^*(s, a).$$

Therefore,

$$\bar{R}^e(s^e, a^e) + \bar{R}^m(s^m, a^m) + \gamma\sum_{s'}\bar{P}(s' \mid s, a)V_\downarrow^*(s') + \left(\frac{\beta^e}{\sqrt{v^e}} + \frac{\beta^m}{\sqrt{v}} + \frac{\beta}{\sqrt{v}}\right) \ge Q_\downarrow^*(s, a). \tag{24}$$

By the union bound over $\mathcal{S}, \mathcal{A}$ and $m_2$, Equation (24) holds for all state-actions until they are visited $m_2$ times, with probability at least $1 - \frac{\delta_2}{2}$. However, instead of the left-hand side of Equation (24), Monitored MBIE-EB follows $\widetilde{Q}_{\text{opt}}$:

$$\widetilde{Q}_{\text{opt}}(s, a) = \bar{R}^e(s^e, a^e) + \bar{R}^m(s^m, a^m) + \gamma\sum_{s'}\bar{P}(s' \mid s, a)\widetilde{V}_{\text{opt}}(s') + \left(\frac{\beta^e}{\sqrt{v^e}} + \frac{\beta^m}{\sqrt{v}} + \frac{\beta}{\sqrt{v}}\right). \tag{25}$$

Therefore, by following the exact induction of Strehl & Littman (2008, Lemma 7), we prove $\widetilde{Q}_{\text{opt}}(s, a) \ge Q_\downarrow^*(s, a)$. Let

$$C = \left(\frac{\beta^e}{\sqrt{v^e}} + \frac{\beta^m}{\sqrt{v}} + \frac{\beta}{\sqrt{v}}\right).$$

Proof by induction is on the value iteration. Let $\widetilde{Q}_{\text{opt}}^i$ be the $i$th step of the value iteration for $(s, a)$. By the optimistic initialization, we have that $\widetilde{Q}_{\text{opt}}^0 \ge Q_\downarrow^*(s, a)$ for all state-action pairs. Now suppose the claim holds for $\widetilde{Q}_{\text{opt}}^i$, we have,

$$\widetilde{Q}_{\text{opt}}^{i+1}(s, a) = \bar{R}^e(s^e, a^e) + \bar{R}^m(s^m, a^m) + \gamma\sum_{s'}\bar{P}(s' \mid s, a)\max_{a'}\widetilde{Q}_{\text{opt}}^i(s', a') + C$$

$$= \bar{R}^e(s^e, a^e) + \bar{R}^m(s^m, a^m) + \gamma\sum_{s'}\bar{P}(s' \mid s, a)\widetilde{V}_{\text{opt}}^i(s') + C$$

$$\ge \bar{R}^e(s^e, a^e) + \bar{R}^m(s^m, a^m) + \gamma\sum_{s'}\bar{P}(s' \mid s, a)V_\downarrow^*(s') + C \qquad\text{(Using induction)}$$

$$\ge Q_\downarrow^*(s, a). \qquad\text{(Using Equation (24))}$$

G.4.2. WHEN $v^{\mathrm{E}} = 0$

If for $(s, a)$, $v^{\mathrm{e}}$ is zero, then Monitored MBIE-EB assigns $-r_{\max}^{\mathrm{e}}$ to $\bar{R}(s^{\mathrm{e}}, a^{\mathrm{e}})$ deterministically. Therefore, the previously random variable $X_1$ in Appendix G.4.1 is deterministically 0. Consequently, Equation (23) take the the form of Equation (26):

$$\mathbb{P}\left(X_2 + X_3 \geq \frac{\beta^{\mathrm{m}}}{\sqrt{v}} + \frac{\beta}{\sqrt{v}}\right) \leq \frac{\delta_2}{3\,|\mathcal{S}|\,|\mathcal{A}|\,m_2}, \tag{26}$$

where $\beta$ and $\beta^{\mathrm{m}}$ are as in Appendix G.4.1. Then, with probability at least $1 - \frac{\delta_2}{3|\mathcal{S}||\mathcal{A}|m_2}$ it must hold that $X_2 + X_3 \leq \left(\frac{\beta^{\mathrm{m}}}{\sqrt{v}} + \frac{\beta}{\sqrt{v}}\right)$, which is equal to

$$\frac{1}{v}\sum_{j=1}^{v}\left(R_j^{\mathrm{m}}(s^{\mathrm{m}}, a^{\mathrm{m}}) + \gamma\bar{V}_{\downarrow}^{*}(S_j')\right) + \left(\frac{\beta^{\mathrm{m}}}{\sqrt{v}} + \frac{\beta}{\sqrt{v}}\right) \geq \mathbb{E}\left[R_1^{\mathrm{m}}(s^{\mathrm{m}}, a^{\mathrm{m}}) + \gamma\bar{V}_{\downarrow}^{*}(S_1')\right] = Q_{\downarrow}^{*}(s, a) - (-r_{\max}^{\mathrm{e}}).$$

Therefore,

$$-r_{\max}^{\mathrm{e}} + \bar{R}^{\mathrm{m}}(s^{\mathrm{m}}, a^{\mathrm{m}}) + \gamma\sum_{s'}\bar{P}(s' \mid s, a)V_{\downarrow}^{*}(s') + \left(\frac{\beta^{\mathrm{m}}}{\sqrt{v}} + \frac{\beta}{\sqrt{v}}\right) \geq Q_{\downarrow}^{*}(s, a). \tag{27}$$

The rest of the proof is identical to the Appendix G.4.1's induction, but with probability at least $1 - \frac{\delta_2}{3}$. Overall, considering Appendices G.4.1 and G.4.2, with probability at least $1 - \frac{\delta_2}{2} - \frac{\delta_2}{3} = 1 - \frac{5\delta_2}{6}$, $\widetilde{Q}_{\mathrm{opt}}(s, a) \geq Q_{\downarrow}^{*}(s, a)$ for all state-actions.

### G.5. Sample Complexity

An algorithm is probably approximately correct in an MDP (PAC-MDP) if for any MDP $M = \langle\mathcal{S}, \mathcal{A}, r, p, \gamma\rangle$ and $\delta, \varepsilon > 0$, it finds an $\varepsilon$-optimal policy in $M$ in time polynomial to $(|\mathcal{S}|, |\mathcal{A}|, \frac{1}{\varepsilon}, \log\frac{1}{\delta}, \frac{1}{1-\gamma}, r_{\max}^{\mathrm{e}})$ (Fiechter, 1994; Kakade, 2003; Szita & Szepesvári, 2010). In Mon-MDPs, partial observability of the environment reward naturally makes any algorithm take more time, as the lower the probability is, the more samples are required to confidently approximate the statistics of the environment reward. As a result, we extend the definition of PAC in MDPs to Mon-MDPs:

**Definition G.1.** An algorithm is PAC-Mon-MDP minimax-optimal if for any Mon-MDP $M = \langle\mathcal{S}, \mathcal{A}, r, p, f^{\mathrm{m}}, \gamma\rangle$ and $\delta, \varepsilon > 0$, it finds an $\varepsilon$-optimal policy in $M_{\downarrow}$, in time polynomial to $(|\mathcal{S}|, |\mathcal{A}|, \frac{1}{\varepsilon}, \log\frac{1}{\delta}, \frac{1}{1-\gamma}, r_{\max}^{\mathrm{e}} + r_{\max}^{\mathrm{m}}, \frac{1}{\rho})$, where $\rho$ is the minimum non-zero probability of observing the environment reward embedded in $f^{\mathrm{m}}$.

Monitored MBIE proceeds in episodes of maximum length $H$ and it computes its action-values at the beginning of each episode. Let $\delta_1 = \delta_2 = \frac{\delta'}{2}, \varepsilon_1 = \varepsilon_2 = \frac{\varepsilon}{2}$. During observe episodes, as was discussed in Appendix G.1, learning transition dynamics is dominant. The rewards are also in $[0, 1]$; hence, we exactly leverage the MBIE-EB's sample complexity for the observe episodes: With probability at least $1 - \frac{\delta'}{4}$ the sample complexity of Monitored MBIE-EB during observe episodes is

$$\widetilde{\mathcal{O}}\left(\frac{m_1\,|\mathcal{S}|\,|\mathcal{A}|\,H}{\varepsilon_1(1-\gamma)}\right) = \widetilde{\mathcal{O}}\left(\frac{|\mathcal{S}|^2\,|\mathcal{A}|\,H}{\varepsilon^3(1-\gamma)^5}\right).$$

It means that in the worst-case scenario in each episode only one unknown state-action is visited with probability at least $\varepsilon_1(1-\gamma)$ and these visits should be repeated $m_1$ times. This specifies the order of $m_3$ and $\kappa^{*}(k)$ in Theorem 3.1:

$$\kappa^{*}(k) = m_3 = \widetilde{\mathcal{O}}\left(\frac{m_1\,|\mathcal{S}|\,|\mathcal{A}|}{\varepsilon_1(1-\gamma)}\right) = \widetilde{\mathcal{O}}\left(\frac{|\mathcal{S}|^2\,|\mathcal{A}|}{\varepsilon^3(1-\gamma)^5}\right).$$

Appendix G.3 showed during optimize episodes If $\rho^{-1} < \mathcal{O}(|\mathcal{S}|)$, the dominant factor in learning models is transition dynamics. Hence, with probability at least $1 - \frac{5\delta'}{12}$, the sample complexity during optimize episodes is equal to

$$\widetilde{\mathcal{O}}\left(\frac{m_2\,|\mathcal{S}|\,|\mathcal{A}|\,H}{\varepsilon_2(1-\gamma)}\right).$$

If $m_2 \geq m_1$, the overall sample complexity (optimize and observe episodes together) is

$$\widetilde{\mathcal{O}}\left(\frac{m_1\,|\mathcal{S}|\,|\mathcal{A}|\,H}{\varepsilon_1(1-\gamma)}\right) = \widetilde{\mathcal{O}}\left(\frac{m_2\,|\mathcal{S}|\,|\mathcal{A}|\,H}{\varepsilon_2(1-\gamma)}\right) = \widetilde{\mathcal{O}}\left(\frac{|\mathcal{S}|^2\,|\mathcal{A}|\,H}{\varepsilon^3(1-\gamma)^5}\right).$$

However, if $\rho^{-1} > \mathcal{O}(|\mathcal{S}|)$, then the dominant factor in learning empirical models is learning the environment reward model. The sample complexity during optimize episodes when $\rho^{-1} > \mathcal{O}(|\mathcal{S}|)$, with probability at least $1 - \frac{5\delta'}{12}$, is

$$\widetilde{\mathcal{O}}\left(\frac{m_2 |\mathcal{S}| |\mathcal{A}| H}{\varepsilon_2 (1 - \gamma)}\right) = \widetilde{\mathcal{O}}\left(\frac{|\mathcal{S}| |\mathcal{A}| H}{\varepsilon^3 (1 - \gamma)^5 \rho}\right).$$

Therefore, we conclude the worst-case scenario is when $\rho^{-1} > \mathcal{O}(|\mathcal{S}|)$ and the final overall sample complexity is dominated by the optimize episodes. This sample complexity holds with probability at least $1 - \frac{5\delta'}{12}$ and is equal to

$$\widetilde{\mathcal{O}}\left(\frac{|\mathcal{S}| |\mathcal{A}| H}{\varepsilon^3 (1 - \gamma)^5 \rho}\right).$$

Defining $\delta = \frac{5\delta'}{12}$ completes the proof. $\square$

## H. Mon-MDPs' Solvability

**Definition H.1** (Parisi et al. (2024b, Definition 1)). Let $M = \langle \mathcal{S}, \mathcal{A}, r, p, f^{\mathrm{m}}\rangle$ be a truthful Mon-MDP. Let $\mathcal{M}$ be the set of all Mon-MDPs that differ from $M$ only in $r^{\mathrm{e}}$. Define $\Pi_M$ to be the set of all policies in $M$ and $\Pi = \bigcup_{M' \in \mathcal{M}} \Pi_{M'}$ to be the set of all Mon-MDPs' policies in $\mathcal{M}$. Further, let $\tau_\ell = \left\{ \left(S_i, A_i, \widehat{R}^{\mathrm{e}}_{i+1}, R^{\mathrm{m}}_{i+1}, S_{i+1}\right)_{i=0}^{\ell} \middle| \pi, M \right\}$ be a trajectory of length $\ell$ in $M$ when following a policy $\pi$, where $\mathbb{E}\left[\widehat{R}^{\mathrm{e}}_{i+1} \middle| \widehat{R}^{\mathrm{e}}_{i+1} \neq \perp, S_i, A_i\right] = r^{\mathrm{e}}(S^{\mathrm{e}}_i, A^{\mathrm{e}}_i)$ almost surely. Let $\mathcal{T}_L = \bigcup_{\mathcal{M} \times \Pi} \left(\cup_{l=0}^{L-1} \tau_\ell\right)$ be the set of all $L$ length trajectories in $\mathcal{M}$. The indistinguishability relation $\mathbb{I}$ between Mon-MDPs $M_1, M_2 \in \mathcal{M}$ is defined:

$$M_1 \mathbb{I} M_2 : \forall L \in \mathbb{N}, \forall \tau \in \mathcal{T}_L, \mathbb{P}\left(\tau | M_1\right) = \mathbb{P}\left(\tau | M_2\right) \quad \text{almost surely.}$$

It follows directly from the definition that the indistinguishability is an equivalence relation:

1. **Reflexive.** Every $M$ is indistinguishable from itself: $M \mathbb{I} M$.

2. **Symmetric.** If $M_1$ is indistinguishable from $M_2$, $M_2$ is also indistinguishable from $M_1$: $M_1 \mathbb{I} M_2 \Leftrightarrow M_1 \mathbb{I} M_2$

3. **Transitive.** If $M_1$ and $M_2$ are indistinguishable, and $M_2$ and $M_3$, so are $M_1$ and $M_3$: $M_1 \mathbb{I} M_2 \wedge M_2 \mathbb{I} M_3 \Rightarrow M_1 \mathbb{I} M_3$.

As an equivalence relation, $\mathbb{I}$ partitions Mon-MDPs into disjoint classes. If $|[M]_{\mathbb{I}}| = 1$, the agent can eventually identify $M$ and its optimal policy, making $M$ solvable. Otherwise, if $|[M]_{\mathbb{I}}| > 1$, $M$ is unsolvable, as it is indistinguishable from at least one Mon-MDP with possibly different optimal policies.

## I. Dependence On $\rho^{-1}$ Is Unimprovable

The main distinction between Theorem 3.1's bound and MBIE-EB's for MDPs is the existence of $\rho^{-1}$. In this section, by showing the existence of $\rho^{-1}$ in a tight lower bound, we conclude the dependence of Theorem 3.1 on $\rho^{-1}$ is tight and essentially unimprovable. Note that lower bounds quantify the difficulty of learning for a given problem for *any algorithm*.

To provide the lower bound, we consider the problem of a stochastic bandit with finitely-many arms (multi-armed bandit for brevity) (Lattimore & Szepesvári, 2020) as a simpler form of sequential decision-making. A multi-armed bandit is a special case of MDPs where the state-space $\mathcal{S}$ is a singleton and the discount factor $\gamma$ equals zero. Mannor & Tsitsiklis (2004) has proved a tight lower bound on the sample complexity of learning in multi-armed bandits. We follow the setup of Mannor & Tsitsiklis (2004) as follows: The agent has $k + 1$ arms (actions). Each arm $a \in [k]$ is associated with a sequence of identically distributed Bernoulli random variables $X_{at}$ with unknown mean $\mu_a$. Here, $X_{at}$ corresponds to the reward obtained the $t$th time arm $a$ is pulled. We assume that random variables $X_{at}$ for $a = 1, \ldots, k + 1$, and $t = 1, \ldots$ are independent. The last arm $a = k + 1$ has a known mean of zero and pulling this arm terminates the interaction. A policy is a mapping that given a history, chooses a particular arm. We only consider policies that are guaranteed to eventually pull arm $k + 1$ with probability one, for every possible vector of means $[\mu_1, \ldots, \mu_k, 0]$ (otherwise the expected number of interaction's steps would be infinite). Given a particular policy and multi-armed Bernoulli bandit, we let $\mathbb{P}$ and $\mathbb{E}$ denote the induced probability measure and the expectation with respect to this measure. This probability measure captures both the

randomness in the arms and the policy. Let $T$ be total number of interaction steps at which the policy chooses arm $k + 1$. Also, let $A_t$ denote the arm chosen at timestep $t$. We say that a policy is $(\epsilon, \delta)$-correct if for every $[\mu_1, \ldots, \mu_k, 0] \in [0, 1]^{k+1}$,

$$\mathbb{P}\left(\mu_{A_{T-1}} > \max_a \mu_a - \epsilon\right) \geq 1 - \delta.$$

**Theorem I.1** (Mannor & Tsitsiklis (2004, Theorem 1)). *There exists positive constants $c_1, c_2, \epsilon_0$, and $\delta_0$, such that for every $k \geq 2, \epsilon \in (0, \epsilon_0)$, and $\delta \in (0, \delta_0)$, and for every $(\epsilon, \delta)$-correct policy, there exists some $[\mu_1, \ldots, \mu_k, 0]$ such that*

$$\mathbb{E}[T - 1] \geq c_1 \frac{k}{\epsilon^2} \ln \frac{c_2}{\delta}.$$

*In particular, $\epsilon_0$ and $\delta_0$ can be taken equal to $0.125$ and $0.25e^{-4}$, respectively.*

In Corollary I.2, we derive a lower bound using Theorem I.1, where each arm's reward is only observed with probability $\rho$.

**Corollary I.2.** *Under the condition of Theorem I.1 with the addition that the reward of each arm is only revealed with probability $0 < \rho < 1$ and with probability $1 - \rho$ the symbol $\perp$ is revealed, then*

$$\mathbb{E}[T - 1] \geq c_1 \frac{k}{\rho\epsilon^2} \ln \frac{c_2}{\delta}.$$

*Proof.* Since we only consider policies that terminates with probability one, then there exists an $n \in \mathbb{N}$ such that $T \leq n$. Let $\widehat{X}_i$ denote the signal received at round $i = 1, 2, \ldots$. We have

$$\mathbb{E}[T] = \mathbb{E}\left[\sum_{t=1}^{n}\sum_{a=1}^{k} \mathbb{1}\{A_t = a\}\right] + 1$$

$$= \sum_{t=1}^{n}\sum_{a=1}^{k} \left(\mathbb{E}[\mathbb{1}\{A_t = a\}]\right) + 1$$

$$= \sum_{t=1}^{n}\sum_{a=1}^{k} \left(\mathbb{E}\left[\mathbb{1}\{A_t = a\}\Big|\widehat{X}_t \neq \perp\right] + \mathbb{E}\left[\mathbb{1}\{A_t = a\}\Big|\widehat{X}_t = \perp\right]\right) + 1$$

$$\geq \sum_{t=1}^{n}\sum_{a=1}^{k} \left(\mathbb{E}\left[\mathbb{1}\{A_t = a\}\Big|\widehat{X}_t \neq \perp\right]\right) + 1$$

$$= \sum_{t=1}^{n}\sum_{a=1}^{k} \left(\frac{\mathbb{E}\left[\mathbb{1}\left\{A_t = a, \widehat{X}_t \neq \perp\right\}\right]}{\mathbb{P}\left(\widehat{X}_t \neq \perp\right)}\right) + 1 \qquad \text{(Conditional expectation's definition)}$$

$$= \frac{1}{\rho}\sum_{t=1}^{n}\sum_{a=1}^{k} \left(\mathbb{E}\left[\mathbb{1}\left\{A_t = a, \widehat{X}_t \neq \perp\right\}\right]\right) + 1$$

$$= \frac{1}{\rho}\mathbb{E}\left[\sum_{t=1}^{n}\sum_{a=1}^{k} \mathbb{1}\left\{A_t = a, \widehat{X}_t \neq \perp\right\}\right] + 1$$

$$= \frac{1}{\rho}c_1 \frac{k}{\epsilon^2} \ln \frac{c_2}{\delta} + 1 \qquad \text{(Theorem I.1)}.$$

Thus

$$\mathbb{E}[T - 1] \geq c_1 \frac{k}{\rho\epsilon^2} \ln \frac{c_2}{\delta}. \qquad \square$$

The existence of $\rho^{-1}$ in the lower bound of Corollary I.2 asserts that the dependence of the Monitor MBIE-EB's sample complexity on $\rho^{-1}$ is tight and unimprovable.

