# OpenReview forum: "Model-Based Exploration in Monitored Markov Decision Processes"
_ICML.cc/2025/Conference — ICML 2025 poster_

### Official Review · Reviewer_TsqN · 2025-03-10

**Overall Recommendation:** 4

**Summary:**

This paper considers model-based reinforcement learning under partially observable rewards. In classical reinforcement learning, the agent always receives a reward upon executing an action in the current state of the underlying MDP. However, in practical scenarios, such as when a human provides intermittent rewards, this assumption may not hold.

The work builds on the recent model of Monitored MDPs, where the agent interacts with both the environment and a monitor (another MDP). The monitor has its own state and action spaces, both observable to the agent. The agent operates in the joint state space, executing joint actions and receiving up to two rewards (monitor and environment), both through the monitor. The crucial difference is that the monitor may hide environment rewards from the agent, though observed rewards are assumed to be truthful.

Due to partial observability of rewards, globally optimal behavior may be unattainable when some rewards are always hidden (unsolvable Mon-MDPs). Thus, the goal is to derive a minimax-optimal policy, ensuring robustness under worst-case assumptions on unobservable rewards.

To address this, the authors extend the MBIE-EB algorithm to Mon-MDPs, the primary challenge being to balance exploration-driven optimism with safety-driven pessimism. They adapt the exploration bonus of MBIE-EB by distinguishing previously observed environment rewards and leveraging KL-UCB to estimate upper confidence bounds on the probability of reward observability. This fosters initial optimism for exploration, which for unobservable rewards shifts to pessimism in the long-run.

The key theoretical result is an MBIE-EB-like sample complexity guarantee for the algorithm to arrive at a near minimax optimal policy, which they prove to be polynomial in parameters corresponding to confidence and proximity to the minimax-optimal expected reward.  This is subsequently evaluated experimentally on a range of Mon-MDP environments, primarily comparing the new MBIE-EB-based approach to existing techniques for Mon-MDPs.

**Claims And Evidence:**

The claims of the paper are well supported. Theoretical claims build on established results from the literature, with proofs given for novel contributions. Practical claims are backed by the experimental evaluation.

**Essential References Not Discussed:**

None that I'm aware of.

**Experimental Designs Or Analyses:**

The benchmark environments, hyperparameters, and experimental setups are well documented and reasonable. The results also appear consistent and well-founded.

**Methods And Evaluation Criteria:**

The considered model class of Mon-MDPs is very recent, therefore, the range of existing benchmark environments is understandably limited. The paper evaluates the proposed approach on a small established benchmark and several seemingly novel synthetic benchmarks. Their choice is reasonable and they cover a range of interesting, though small-scale setups. Since the algorithm is model- and counting-based, all environments are grid worlds.

**Other Comments Or Suggestions:**

- Line 126: Footnote 1 should probably be Footnote 3.
- Line 174: r_max should probably be r_min.
- Line 170: The definition of the equivalence class [M]_I for Mon-MDPs under reward observability is not quite self-contained here and should probably include a little clarification on state and action spaces being fixed as per [Parisi et al., 2024b] . The description “the set of all Mon-MDPs that the agent cannot distinguish” confused me for a second, since this may also include much larger but, e.g., bisimilar Mon-MDPs, also rendering the set not a singleton for solvable Mon-MDPs.

**Other Strengths And Weaknesses:**

# Strengths

The paper is overall very well written and easy to follow. The authors do a good job in keeping the paper self-contained and motivating their contributions. The proposed algorithm is simple and elegant, yet effective as both formally proven in their main theoretical results and practically validated in the experimental evaluation, primarily comparing it to the existing Directed-E^2 algorithm for Mon-MDPs.
The theoretical results are compelling, extending MBIE-EB-like guarantees from standard MDPs to the more complex model of Mon-MDPs. The evaluation demonstrates a clear advantage of the new algorithm over existing methods, especially in unsolvable Mon-MDPs where some rewards are never observable, for which, e.g., the Directed-E^2 algorithm seems to struggle.
# Weaknesses
I am wondering about the practical applicability of the Mon-MDP setup. The authors motivate the model with, for example, a human in the loop who provides rewards but may not always be present, thus making some rewards unobservable. While this makes sense conceptually, it also seems like a fairly coarse-grained approach in which all rewards are either observable or not during certain time periods, not emitting much more exploitable structure.
At the same time, the model class assumes full observability of the monitor’s state space by the agent, and it’s not entirely clear how this would extend to, for example, a human-in-the-loop setting or other complex scenarios beyond a simple present/not-present setup. The monitors the authors consider are mainly random, small “button-like” states (on/off), or cases where the agent must determine which part of the monitor to consult. I’m wondering to what extent this assumption of full observability restricts potential applications of this theoretically compelling framework, and whether it might be relaxed into a partially observable state and reward setting (like Mon-POMDPs?). However, I also recognize that this may not be a shortcoming of the current paper, since it does not introduce the model class itself but proposes a new algorithm that improves upon existing ones.
The experimental evaluation is broad and covers a range of grid-worlds with diverse features and complications, all tested with different monitor designs. However, these environments are relatively small, with the largest being a 6x6 2D grid-world. While smaller setups are typical in model-based approaches, it would be interesting to see how this method scales to larger or higher-dimensional domains.

**Questions For Authors:**

1. To what extent does the assumption of a fully observable monitor state space restrict the applicability of this approach in real-world scenarios? Are there practical examples where this assumption is clearly justified?
2. Would an extension of this to partially observable states (i.e., unobservable monitor states) be possible (like Mon-POMDPs)?
3. How does this approach scale to larger environments in higher dimensions? What are the limiting factors, e.g., compared to plain MBIE-EB?

**Relation To Broader Scientific Literature:**

The paper builds and extends on the recent literature on Mon-MDPs introduced by [Parisi et al., 2024b] ( AAMAS 2024), and partially observable rewards [Parisi et al., 2024a] (NeurIPS 2024). The authors extend the MBIE-EB algorithm from [Strehl and Littman, 2008] to finite-state Mon-MDPs, also leveraging upper confidence bounds as per [Garivier and Cappé, 2011] for the estimation of the chance to observe rewards. In combination, this results in a model-based RL algorithm for Mon-MDPs which they compare to the Directed-E^2 algorithm from [Parisi et al., 2024a].

**Theoretical Claims:**

The main theoretical claim builds on established results from the literature. I reviewed the corresponding proofs in the appendix and found no inaccuracies or gaps.

---

> ### Author Rebuttal · Authors · 2025-03-31
>
> We thank Reviewer TsqN for providing comprehensive and in-depth feedback!  We are glad you found the paper well-written and easy to follow. Here, we try to address the points mentioned in order:
>
> >  I’m wondering to what extent this assumption of full observability restricts potential applications of this theoretically compelling framework, and whether it might be relaxed into a partially observable state and reward setting (like Mon-POMDPs?)
>
> As you have mentioned, the introduction of the Mon-MDPs is not a contribution of our work, but we find it useful to share our thoughts around your insightful scrutiny around the applicability of Mon-MDPs.
>
> We agree that grid-world domains do not represent the real-world applicability of most algorithms. Our choice to run experiments on these domains is because the Mon-MDP framework is relatively new and controlled small-scale experiments help shed light on the understanding of various aspects of this framework. Following the original Mon-MDP paper [Parisi 2024b], we have assumed that the agent receives a joint state from the environment and the monitor. Since these are both MDPs, they could likely be modeled by POMDPs instead. In such a (novel) framework, we imagine the agent would receive the joint observation and techniques from the POMDP literature could be adapted in future work.
>
>
> >  The definition of the equivalence class [M]_I for Mon-MDPs under reward observability is not quite self-contained here and should probably include a little clarification on state and action spaces being fixed as per [Parisi et al., 2024b] . The description “the set of all Mon-MDPs that the agent cannot distinguish” confused me for a second, since this may also include much larger but, e.g., bisimilar Mon-MDPs, also rendering the set not a singleton for solvable Mon-MDPs.
>
> As you have pointed out, the definition is given by [Parisi et al 2024b]. But we revisit it in the appendix of the final version and further clarify the concept.
>
>
> >To what extent does the assumption of a fully observable monitor state space restrict the applicability of this approach in real-world scenarios? Are there practical examples where this assumption is clearly justified?
>
> If the state of the monitor and the environment are fully observable to the agent, then Mon-MDPs would suffice.  We feel that many monitored settings match the fully observable criteria.  Often the RL practitioner is instrumenting the system that provides rewards, and so the full state is more easily known and possibly provided to the learning agent (e.g., the status of sensors that measure reward, or their locations in the world and under what conditions they provide signals).  Human monitors, though, might be a good example that would not fit the fully observable setting as they typically would have internal state that could not be provided to the agent.
>
> > Would an extension of this to partially observable states (i.e., unobservable monitor states) be possible (like Mon-POMDPs)?
>
>  Likely - please see our answer above.
>
> > How does this approach scale to larger environments in higher dimensions? What are the limiting factors, e.g., compared to plain MBIE-EB?
>
> The biggest limiting factor is the direct dependency on the counts of visiting state-action pairs or observing the environment rewards (which applies to other tabular methods, including MBIE-EB). As pointed out in the paper in Discussion section Iine 416, using pseudocounts could ameliorate this shortcoming to some extent, which have been applied to improve exploration with Deep RL methods [Bellemare et al., 2016; Tang et al., 2017]. The other limitation (distinct from MBIE-EB) is the computation time of the KL_UCB term, which does not have a closed-form. However, this term can be replaced with a typical $\frac{\alpha}{\sqrt{N(s, a)}}, \alpha > 0$ bonus. This new bonus again can be extended by pseudocounts.

---

> > ### Comment · Reviewer_TsqN · 2025-04-03
> >
> > I thank the authors for comprehensively addressing my questions and concerns. Having read their response and the other reviews, I find that the presented work is a nice contribution to the so far rather niche, but as other reviewers also point out, important field of Mon-MDPs, which might offer promising potential for future developments in RL under partially observable rewards. I think the authors should make the motivation and concrete examples of applications that fit into the framework of the current work more prominent and clear in the paper, as they did in their response. In particular, it would be helpful to also highlight what is currently not covered by the present framework, such as human-in-the-loop setups with unobservable internal state of the monitor. I believe that the direction of partially observable monitor states would be a very interesting avenue for future work in this line of research.
> >
> > Since the paper is very well written and comprehensive and advances the field of Mon-MDPs and partially observable rewards, I decided to raise my score and recommend acceptance, although I strongly encourage the authors to address the comments provided in the reviews.

---

### Official Review · Reviewer_gjFL · 2025-03-10

**Overall Recommendation:** 3

**Summary:**

The current paper consider the task of bounding the sample complexity of exploration in monitored MDP where the reward of the environment is observed only for some of the state action pairs and where the minimum non zero probability of observing an environment reward is lower bounded by $\rho$.

Interesting the authors bound the sample complexity of exploration aslo for Non solvable Mon-MDP using as comparator not the optimal policy but the minmax policy which is optimal in the wrost case MDP. That is, the one that has minimum reward in the state action pairs for which the environment never reveals the reward function.

### After rebuttal

My opinion remains positive.
I think that the lower bound will improve the paper. I hope you can really include it in the final version.

**Claims And Evidence:**

Yes, there is clear and convincing evidence.

**Essential References Not Discussed:**

See above

**Experimental Designs Or Analyses:**

The experiments are complete. They are limited to tabular tasks but it makes sense to run experiment exactly in the same setting for which the theory applies.

**Methods And Evaluation Criteria:**

Yes, the proposed experiments and theory make sense for the considered problem.

An additional theoretical result that the author could try to obtain is the regret from the initial distribution ( allowing the agent to reset from it).
In this way the author could argue that they can recover a policy almost as good as the minmax policy in solving the Mon-MDP. The current guarantees only say that the recovered policy is as good as the minmax one for states that are visited by the learner after a long enough number of steps.
I think that this is an interesting future direction that the authors could pursue.

**Other Comments Or Suggestions:**

I think that the author should add a pseudocode of the algorithm.

**Other Strengths And Weaknesses:**

The algorithmic derivation is very well explained. I particularly liked the steps explaining why plain UCB approaches would fail for excessive optimism and why assigning $r_\min$ to unobserved state action pairs would fail due to excessive pessimism.

**Questions For Authors:**

Have you tried to prove a lowe rbound to show that the dependence on $\rho^{-1}$ is necessary ?

**Relation To Broader Scientific Literature:**

I think that the related literature is well covered.
However, I think that the author should motivate the metric that they bound in their theoretical guarantees a bit better.
In particular, they should say that this quantity is known as sample complexity of exploration and it was first studied in Kakade's PhD thesis.

https://homes.cs.washington.edu/~sham/papers/thesis/sham_thesis.pdf Theorem 8.3.2

Also there are several follow up works looking at the same metric

https://proceedings.mlr.press/v19/szita11a/szita11a.pdf Definition 5.1

Lattimore & Hutter PAC Bounds for Discounted MDPs

I think the authors should add a discussion motivating this metric and referring to these previous works.

**Theoretical Claims:**

The proofs seem correct. I had a quick look at all of them.

---

> ### Author Rebuttal · Authors · 2025-03-31
>
> We thank Reviewer gjFL for providing valuable feedback! We are glad you found the algorithmic derivation very well explained. Here, we try to address the points mentioned in order:
>
> > An additional theoretical result that the author could try to obtain is the regret from the initial distribution ( allowing the agent to reset from it). In this way the author could argue that they can recover a policy almost as good as the minmax policy in solving the Mon-MDP. The current guarantees only say that the recovered policy is as good as the minmax one for states that are visited by the learner after a long enough number of steps. I think that this is an interesting future direction that the authors could pursue.
>
> We thank the author for the suggestion! We agree that would be an interesting future direction.
>
> > I think the authors should add a discussion motivating this metric and referring to these previous works.
>
> We agree.  We’ll cite the suggested works and elaborate more on the motivation behind sample complexity in the final version of the paper.
>
> > I think that the author should add a pseudocode of the algorithm.
>
> We’ll add pseudocode to the appendix in the final version of the paper.
>
> > Have you tried to prove a lower bound to show that the dependence on $\rho^{-1}$ is necessary?
>
> We felt the dependence was indeed necessary, but hadn’t thought about making the argument formal until your question.  One can construct hard examples from simple single-state Mon-MDPs (embodying a bandit problem), where arm rewards are only observable with probability $\rho$.  Intuitively, since it will take $O(1/\rho)$ pulls to observe any arm’s reward, this should give exactly this lower bound.  We will aim to make this formal (building off of Mannor and Tsitsiklis’s (2004) bandit PAC-bounds) in the final version.

---

> > ### Comment · Reviewer_gjFL · 2025-04-02
> >
> > Thanks for your answer!
> >
> > I think that the lower bound will improve the paper. I hope you can really include it in the final version.
> >
> > Best,
> > reviewer

---

> > > ### Author Response · Authors · 2025-04-04
> > >
> > > Building off of Mannor and Tsitsiklis’s (2004), now we have done the derivation and shown the necessary existence of  $1 / \rho$. We will add this finding to the final version.

---

### Official Review · Reviewer_fk7j · 2025-03-12

**Overall Recommendation:** 4

**Summary:**

This work extends model-based interval estimation with exploration bonus (MBIE-EB), a well-known model-based exploration algorithm for MDPs with PAC guarantees, to the monitored MDP (Mon-MDP).

The monitored MDP relaxes the assumption that the reward is observed at every time step and, instead, let this be determined by a monitor.
This monitor can be, for example, a human giving rewards to the agent (but only if they are present) or depend on an unreliable / expensive sensor.
This monitor itself is a state-full (Markovian) entity function that observes the true environment and provides a "monitor" reward and an environment reward.
The goal is to optimize for the true environment return plus the monitor reward.

MBIE-EB works on constructing optimistic MDPs to ensure the problem is fully explored, based on counting visits.
The extension proposed here has two parts.
First, the optimistic counting is also applied to the transition and reward function of the monitor.
Second, to good worst-case performance when rewards of certain state-action pairs are never observable, the agent becomes pessimistic after a certain horizon (hyper parameter).

This method is compared to directed exploration-exploitation (Directed-E2) on several benchmarks reported in Directed-E2's paper.
There are no other baselines, which is understandable given the lack of SOTA due to the recency of Mon-MDP's definition, but the experimental section is insightful and convincing.
Lastly, the method comes with sample complexity, which seems to be based on MBIE-EB's original proof (techniques), but contains some novel components.

## update after rebuttal

The rebuttal period has not changed my mind regarding the (positive) score.

**Claims And Evidence:**

= Worst-case performance

The method is claimed to learn policies for worst-case Mon-MDP scenarios (of never observing rewards in some state-action pairs).
This is well tackled both intuitively when introducing the method, as well as in empirical studies on convincing domains.

= Sample complexity guarantees.

It is claimed to have the first sample-complexity guarantees and the proof _looks_ good.

**Essential References Not Discussed:**

Nothing came to my mind.

**Experimental Designs Or Analyses:**

The usual experiments on performance over time, which is fine.

**Methods And Evaluation Criteria:**

See summary, the method is evaluated on Mon-MDP benchmarks where they show their proposed method reliably outperforms the SOTA (for which the domains were designed).
It would have been nice to see some non Mon-MDP algorithms performance, especially on MDPs that are less obviously designed with these Mon-MDP algorithms in mind.
Similarly, some ablation studies on parameter robustness and different versions of the proposed method would have been insightful and helpful.

**Other Comments Or Suggestions:**

It would have been nice to introduce / describe KL-UCB in more depth, at least define it mathematically.

**Other Strengths And Weaknesses:**

Nothing comes to mind.

**Questions For Authors:**

Nothing comes to mind.

**Relation To Broader Scientific Literature:**

This work is specifically targeting Mon-MDPs, which is a rather niche (but I can see important) problem setting.
The contributions are unlikely to have broad influence in other fields as, instead, it is applying (non-trivially) an existing algorithm / proof to their use-case.

**Theoretical Claims:**

The method is shown to converge within finite samples, depending on parameters inherited form MBIE-EB and the minimum non-zero probability that a reward is seen `p`.
In particular, the sample complexity is `O(... 1/p)`.

---

> ### Author Rebuttal · Authors · 2025-03-31
>
> We thank Reviewer fk7j for providing valuable feedback! Here, we try to address the points mentioned in order:
>
> > It would have been nice to see some non Mon-MDP algorithms performance, especially on MDPs that are less obviously designed with these Mon-MDP algorithms in mind. Similarly, some ablation studies on parameter robustness and different versions of the proposed method would have been insightful and helpful.
>
>  In Figure 9 in Appendix B.3 the first column shows the performance on traditional MDPs. Parisi et al. (2024a) showed the superior performance of of Directed-$E^2$ against conventional exploration strategies such as $\varepsilon$-greedy, optimistic initialization, $\varepsilon$-greedy with count bonuses, $\varepsilon$-greedy with UCB bonus and $\varepsilon$-greedy with [long-term UCB bonus](https://www.mdpi.com/1999-4893/15/3/81) [Parisi et al., 2022]. Our conclusion is that when our proposed algorithm, Monitored MBIE-EB, outperformed Directed-E$^2$, it simultaneously outperformed five other baselines as well.  We will make this more explicit in the final version.
>
> We have run ablation studies on the significance of pessimism and observe episodes in which the agent only seeks observability of the environment reward. The findings show that both are needed to achieve the results in the paper. We’ll add the ablation experiments to the appendix in the final version.
>
> > It would have been nice to introduce / describe KL-UCB in more depth, at least define it mathematically.
>
> We have given the explicit formula of KL-UCB in Appendix E, page 26, line 1383. We will add a footnote to the main body of the paper with the formula and reference to the appendix.

---

### Official Review · Reviewer_VJG2 · 2025-03-14

**Overall Recommendation:** 3

**Summary:**

In this paper, authors develop a model-based interval estimation algorithm for Monitored MDPs. Authors prove typical desired properties about this algorithm, and compare its performance on a suite of 24 benchmark environments to the previous SOTA for Mon-MDPs called $E^2$. The results show notable improvements.

**Claims And Evidence:**

Claims:
- The new model-based algorithm can fully exploit the problem structure, take advantage of a known monitor, have worst-case guarantees for unsolvable Mon-MDPs even without specific initialization, and have non-asymptotic proofs of convergence.
	- Evidence: Theory work in Section 3 with proofs in the Appendix.
- MBIE-EB performs better than $E^2$ with random initialization:
	- Evidence: Appendix B.4 states that with random initialization, essentially no learning occurs, but I think a Figure showing this and comparing the performance to MBIE-EB would be useful.
- The new algorithm outperforms Mon-MDP baselines such as $E^2$.
	- Evidence: Figure 6 Plots.

**Essential References Not Discussed:**

Unknown

**Experimental Designs Or Analyses:**

Authors provide 95% confidence intervals, and hyperparameters are clearly stated in Appendix C.

**Methods And Evaluation Criteria:**

Authors compare against a suite of 24 benchmark grid environments such as "River Swim". These environments seem appropriate as a first test of this method, and their method outperforms the baseline, $E^2$. I wonder if there are any other relevant baselines that should be included?

They suggest future work such as pseudocounts that would expand the applicability of the approach to non-enumerable state spaces. It seems reasonable for this to be left to future work rather than included in this paper.

**Other Comments Or Suggestions:**

Figure 6: It looks to me like 4/24 of the environments have on par performance, not 2/24.

**Other Strengths And Weaknesses:**

The main weakness of the paper, in my opinion, is that I am not sure how widely used Mon-MDPs are. Within domain, the algorithm suggested in the paper is certainly good, but without further knowledge of this literature, I am not sure what other related extended MDP formulations have similar results known.

**Questions For Authors:**

Not at this time.

**Relation To Broader Scientific Literature:**

This paper extends the classic MBIE algorithm to Mon-MDPs.

**Theoretical Claims:**

No.

---

> ### Author Rebuttal · Authors · 2025-03-31
>
> We thank Reviewer VJG2 for providing valuable feedback! Here, we try to address the points mentioned in order:
>
> > Appendix B.4 states that with random initialization, essentially no learning occurs, but I think a Figure showing this and comparing the performance to MBIE-EB would be useful.
>
> Figure 4.b and Figure 4.c show the lack of Directed-$E^2$'s learning and comparison with Monitored MBIE-EB. We will make a reference to these results in that section of the Appendix.
>
> > Figure 6: It looks to me like 4/24 of the environments have on par performance, not 2/24.
>
> We agree with the reviewer that from Figure 6 alone, 4 of the environments look to have similar performance.  Due to the scaling of the y-axis, the final performance on two of the plots are difficult to evaluate. You can find rescaled results in Figure 9 in Appendix B.3 where its clear 2/24 have on-par performance, which is the basis for the claim.  We will update this figure.

---

### Decision · Program_Chairs · 2025-05-01

**Decision:**

Accept (poster)

**Comment:**

The authors did a good job of addressing the technical concerns raised by the reviewers. The only unaddressed concern(s) could be paraphrased as: Are the authors really solving a problem worth solving? It's disappointing that this issue was raised by multiple reviewers, yet the authors didn't have a compelling answer.